Corrected: Author correction

# Intestinal microbiome adjusts the innate immune setpoint during colonization through negative regulation of MyD88

Bjørn E. V. Koch [1,2], Shuxin Yang[1,3], Gerda Lamers[1], Jens Stougaard [2] & Herman P. Spaink [1]

Host pathways mediating changes in immune states elicited by intestinal microbial colonization are incompletely characterized. Here we describe alterations of the host immune state induced by colonization of germ-free zebrafish larvae with an intestinal microbial community or single bacterial species. We show that microbiota-induced changes in intestinal leukocyte subsets and whole-body host gene expression are dependent on the innate immune adaptor gene *myd88*. Similar patterns of gene expression are elicited by colonization with conventional microbiome, as well as mono-colonization with two different zebrafish commensal bacterial strains. By studying loss-of-function *myd88* mutants, we find that colonization suppresses Myd88 at the mRNA level. Tlr2 is essential for microbiota-induced effects on *myd88* transcription and intestinal immune cell composition.

[1] Institute of Biology, Leiden University, 2333 BE Leiden, The Netherlands. [2] Department of Molecular Biology and Genetics, Aarhus University, 8000 Aarhus C, Denmark. [3] Present address: Center for Synthetic Biology Engineering Research, Shenzhen Institutes of Advanced Technology, Chinese Academy of Sciences, 518055 Shenzhen, China. Correspondence and requests for materials should be addressed to H.P.S. (email: h.p.spaink@biology.leidenuniv.nl)

The microbial communities residing in the intestine comprise a vast and diverse assembly of species that are collectively known as the gut microbiome. Colonization usually occurs at birth[1] and marks the transition of the intestine from virtual sterility into the major site of interaction with the microbiome. Several recent landmark discoveries have clarified the central importance of the microbiome in nutrient uptake, maintenance of energy homeostasis and control of inflammatory signaling[2–4]. Studies of children delivered by cesarean section versus vaginal delivery have shown persistent effects of initial colonization in the host[5,6]. Aberrant microbial colonization is suggested to explain the correlations between cesarean delivery and the elevated risk of developing a wide range of diseases, relating particularly to the mucosal immune regulation[7]. However, the underlying molecular pathways conveying the signaling and signal exchange involved in the establishment of the interdependent relationship between host and microbiome are not understood.

Several animal studies point to the toll-like receptor (TLR) pathway as central in many aspects of host sensing of the microbiome[8–11]. TLR2 has attracted interest in this regard for several reasons. TLR2 mediated microbial pattern recognition has been shown to be important for facilitating tolerance to commensal microbial colonization[12], induction of mucin secretion[13] and protection of intestinal barrier integrity in induced inflammation models[14,15]. With the exception of TLR3, all TLRs can signal through the central intracellular adapter Myd88[16–18]. Several observations support the importance of TLR signaling to host intestinal immune homeostasis, as mouse mutants of specific TLRs have been found to develop inflammatory intestinal conditions without pathogenic challenge[19,20]. However, seemingly conflicting observations have been reported as another recent study found improved intestinal immune responses to a high fat diet in a Myd88 deficient mouse mutant[21]. Clearly there are gaps in our knowledge, and direct comparisons between different studies should be done with caution. Some of the apparent discrepancies regarding the inflammatory properties of TLR signaling are likely caused by differences in experimental design. Microbial presence in the intestine can be detected by various mechanisms and TLR signaling is only one of them. The influence of food intake and microbial metabolism of ingested organic material, particularly short-chain fatty acids, on the overall outcome makes it a complex subject to investigate without confounding influences of diet and secondary metabolites (see ref.[22] for a review). Model systems where the complexity of the system can be controlled and manipulated easily are therefore necessary in order to dissect the contribution of single microbial species or conventional microbial community colonization.

Here we take advantage of the rapid development of zebrafish embryos to study host microbe interactions of scalable complexity. Comparisons between zebrafish and mouse model systems have demonstrated the translational value of the zebrafish model[23]. Utilizing fluorescent reporter fish lines, immunostaining, and genetic mutants we characterize the intestinal innate immune response to bacterial colonization and investigate the role of Myd88 in mediating these effects. Using RNAseq we characterize and compare the effects of colonization at different levels of microbial community complexity, and in the presence or absence of a functional Myd88 encoding gene. Our data reveal aspects of *myd88* transcription controlled by microbial colonization through Tlr2 mediated sensing.

## Results

### Myd88-dependent gut immune cell responses to colonization.
We wanted to investigate how, and to what extent, Myd88 signaling mediates the effects of colonization, especially on the mucosal immune status of the host. To this end we generated germ-free and conventionalized zebrafish larval groups according to established protocols[24]. Fertilized embryos were surface sterilized, maintained germ-free or conventionalized at day 3 and analyzed 5 days post fertilization (DPF) (Fig. 1a). Within these groups we utilized anti-L-plastin, a leukocyte specific antibody[25], to quantify the number of innate immune cells present in the intestinal epithelial cell layer in germ-free versus conventionalized larvae[26]. The results showed significantly elevated numbers of L-plastin positive cells in the distalmost part of the gut of *myd88* +/+ germ-free larvae, but not in *myd88* deficient mutant zebrafish larvae (Fig. 1b, c). This Myd88 dependent elevation of leukocyte presence was also evident in full intestines excised from L-plastin stained germ-free larvae 5 DPF (Supplementary Fig. 1a and Supplementary Fig. 2). Previous observations in zebrafish larvae have found neutrophil presence in the intestine to be lowest in germ-free larvae as compared to colonized groups at 6 DPF[9,27]. As L-plastin, the antigen targeted in our experiments, is a general leukocyte marker, this approach cannot distinguish macrophages from neutrophils. Therefore, we performed the same experiment in the fluorescent neutrophil reporter line *Tg(Mpx:GFP)*[28]. Our results showed a significantly elevated neutrophil infiltration in conventionalized embryos compared to germ-free, and again the regulation was dependent on Myd88 (Fig. 1d, e). Thus our results are consistent with previous observations from similar zebrafish studies[9,27], and indicate that our L-plastin counts reflect elevated macrophage infiltration in germ-free compared to conventionalized larvae. Examination of total numbers of leukocytes (L-plastin positive cells) revealed no significant alteration of overall leukocyte presence in the larvae (Supplementary Fig. 1b, c). To verify the observation in an independent fashion we utilized a double fluorescent line generated by crossing the *Tg(Mpx:GFP)* line with a macrophage reporter line *Tg(Mpeg:mCherryF)*[29], and found that indeed the observed L-plastin elevation was the reflection of an elevated macrophage presence in germ-free intestines (Supplementary Fig. 1 D). Overall these observations show an elevated macrophage presence in the intestines of germ-free embryos, which is alleviated by myd88-mediated signaling upon intestinal microbial colonization.

### Host transcriptome response to microbes is not inflammatory.
Our analyses of leukocytes in the intestines indicate an apparent Myd88-dependent immune reaction to conventionalization of the larvae leading to diminished macrophage presence and an elevated neutrophil presence. To investigate the transcriptional basis of the response and examine the broader systemic impact of microbial presence on the host transcriptome, we performed RNAseq analysis based on total RNA extracts from germ-free versus conventionalized 5 DPF wildtype zebrafish larvae. Analysis of the data identified several transcripts which have previously been described as sensitive to colonization, such as *angptl4* (also known as *fiaf* ) and *nr1d1* (also known as *RevErba*), downregulated by colonization. This confirms that our results are in line with similar studies from zebrafish[8,10] and mice[2,30].

Applying a false discovery rate adjusted *P*-value of 0.05 and a minimal fold change of 1.3 as significance cut-off, we found 257 transcripts that were differentially expressed between germ-free and conventionalized conditions. The majority of the transcripts, 184 of the 257, were suppressed by colonization (Supplementary Fig. 5 and Supplementary Table 1). The upregulated genes were dominated by two groups of transcripts involved in intestinal immune regulation; a group of opsonizing glycoproteins of a gene family with the description "Pancreatic Secretory Granule Membrane Major Glycoprotein GP2", as well as several mucins.

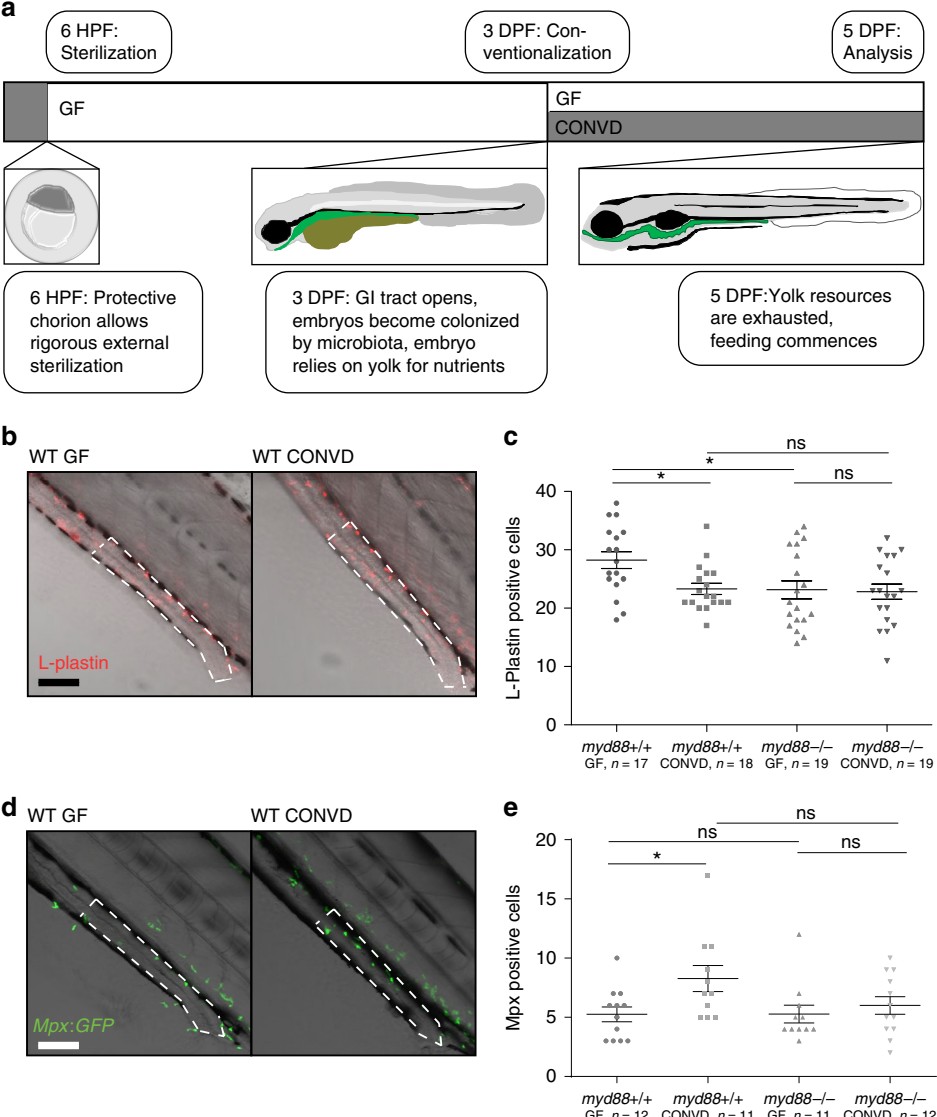

**Fig. 1** Changes to immune cell composition in the gut is Myd88 dependent. **a** Schematic representation of the developmental features of early zebrafish larvae which form the foundation of this experimental approach. **b** Representative images of distal intestines of germ-free (GF) and conventionalized (CONVD) wildtype (WT) larvae following L-plastin staining. Composite images of bright-field and L-plastin signal from confocal maximum intensity Z-projection in red. **c** WT larvae under germ-free conditions exhibit a significantly elevated leukocyte presence in the distal intestine, compared to conventionalized. This elevated leukocyte presence was no longer detectable under the same conditions in *myd88* deficient larvae. Figure is representative of three independent replications. **d** Representative live microscopy images of distal intestine of GF and CONVD larvae of the *Mpx:GFP* reporter zebrafish line. Bright-field overlaid with confocal maximum intensity Z-projection in green. **e** Germ-free larvae exhibit significantly reduced neutrophil infiltration in the distal intestine compared to conventionalized larvae. No significant difference was observed in *myd88* deficient backgrounds. Figure is made from pooled data from three biological replicates. **b**, **d** an area of intestine extending 4 somites proximal to the cloaca, representing the area of counting, is outlined in white dashed lines. scale bars represent 100 μm. **c**, **e** each data point represent cell counts from one larvae based on confocal z-stacks acquired at 20 times magnification. Error bars represent standard error of the mean. *$P \leq 0.05$ by two-way ANOVA with Bonferroni correction for multiple comparisons

Within the group of genes suppressed by colonization were several genes known to be induced by *myd88*-dependent TLR signaling, including *myd88* itself. Additionally, several members of the activating protein-1 (AP1) transcription factors of the *fos*, *jun*, and *atf* families and the signal transducer suppressor of cytokine signaling 3a (*socs3a*) which are known to be regulated in a Myd88 dependent manner[31] were suppressed by colonization. Also, the genes encoding CCAAT/enhancer binding protein beta and delta (*cebpb* and *cebpd*), transcription factors that have been shown to be induced by, and involved in, Myd88 and TLR mediated signaling[32], were suppressed in conventionalized larvae

(Fig. 2a). Thus, the transcriptional profile of the conventionalized group, which must be expected to encounter the most TLR ligands, shows a suppression of Myd88 dependent signaling components of the AP1 transcription complex and ccaat/enhancer binding protein family. Interestingly, NF-κB dependent transcripts such as serum amyloid A (*saa*) and NF-κB inhibitor alpha a (*nfkbiaa*)[10] and inflammatory cytokines such as *il1b*, *tnfa*, and *il6*, characteristic of TLR-stimulated Myd88-dependent signaling in infectious disease, did not exhibit significant transcriptional regulation (Fig. 2a). This observation is consistent with the results of a similar study in zebrafish larvae, which found

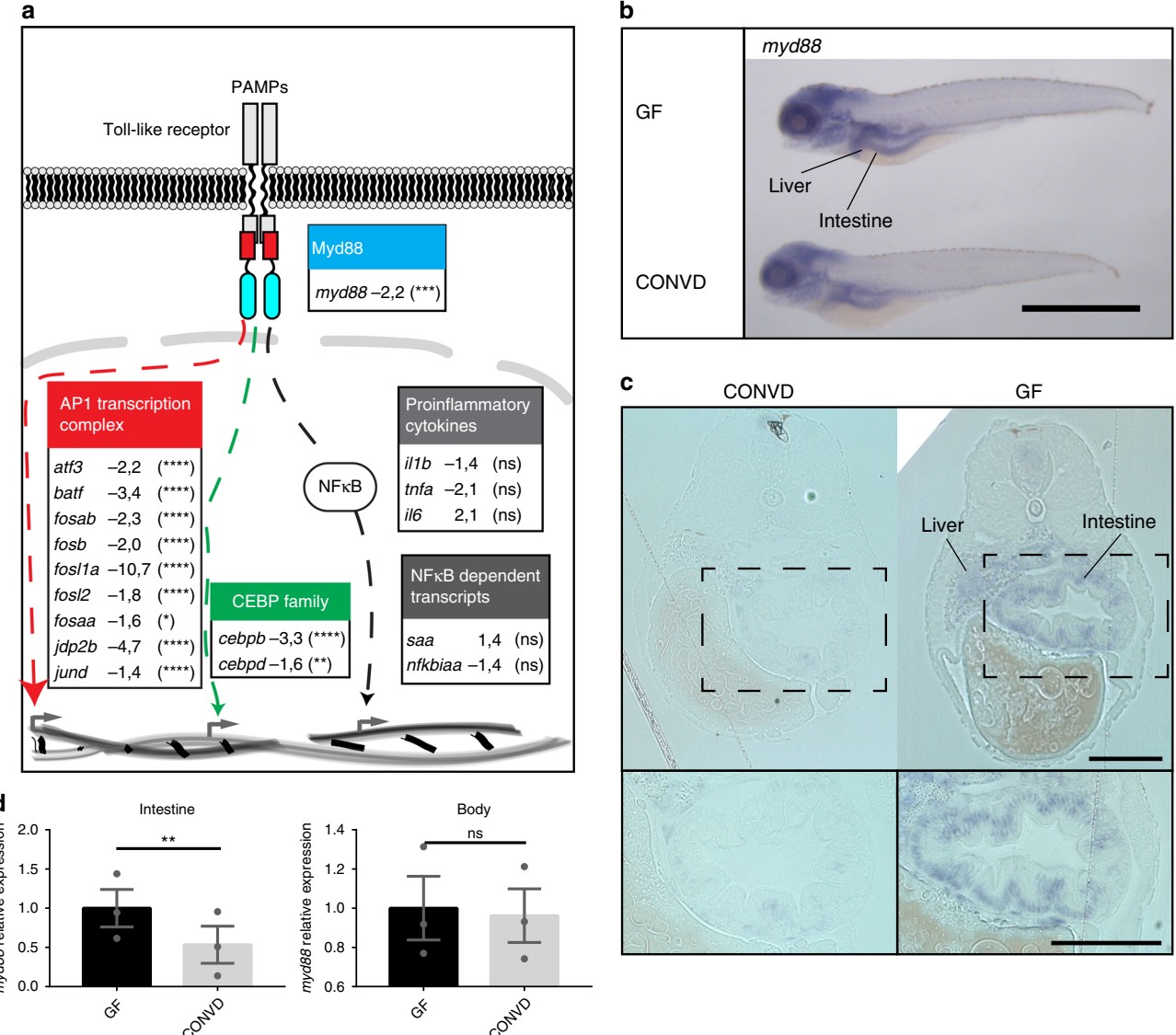

**Fig. 2** Microbes transcriptionally suppress Myd88 and downstream signaling components. **a** Graphic representation of the canonical TLR stimulated Myd88 dependent transcriptional signaling though the AP1 transcription complex (red), the CCAAT/enhancer binding protein family (green) or the NF-κB nuclear translocation (gray), along with associated gene names and fold-change values in the conventionalized group relative to germ-free. Nine transcription factors of the AP1 transcription complex and two members of the CCAAT/enhancer binding protein (C/EBP) family exhibited significant transcriptional suppression upon colonization. The activation of NF-κB is not readily observable by transcriptomics, however neither the NF-κB dependent transcripts serum amyloid A (saa) or NF-κB inhibitor Alpha a (nfkbiaa) or the proinflammatory cytokines normally associated with NF-κB activation were significantly regulated. It should be noted that the transcriptional foldchanges are derived from whole-body transcriptomics and that it cannot be concluded that the transcriptional changes represented here all take place in the same cells, even though they are part of the same regulatory pathway. Statistical evaluations represent the Benjamini-Hochberg adjusted P-values of the RNAseq data comparing conventionalized embryos to germ-free controls. *$P \leq$ 0.05; **$P \leq$ 0.01; ***$P \leq$ 0.001; ****$P \leq$ 0.0001; ns = not significant. **b** Whole-mount in situ hybridization reveals that myd88 is expressed primarily in the intestine and liver in 5 DPF embryos. Scalebar represents 1 mm. **c** 2 μm transverse sections of plastic embedded in situ hybridization of myd88 comparing the pattern in a germ-free and a conventionalized embryo. Scalebar represents 100 μm. **d** qPCR analysis comparing the relative expression levels of myd88 in germ-free versus conventionalized 5 DPF embryos in intestines versus body tissues, (mean ± s.e.m., n = 3 biological replicates, 30 embryos per group), **$P \leq$ 0,01; by Student's t-test

an initial rise in il1b transcription in the immediate hours after colonization, followed by normalization after 2 days[18]. The transcriptomics data sets were based on whole-embryo RNA extracts, and do not provide any information about the tissue(s) in which the regulation takes place. To assess tissue-specific expression levels, we performed whole mount in situ hybridization of three transcripts, myd88, fosl1a, and cebpb. The results showed that transcriptional changes were localized primarily in the gastrointestinal tract and liver (Fig. 2b, c and Supplementary Fig. 3 and 4c). The intestinal regulation of these transcripts was further validated by tissue specific qPCR assessment (Fig. 2d and Supplementary Fig. 4b). The tissue specific qPCR approach further validated that il1b is also not significantly regulated in the intestine (Supplementary Fig. 4b).

In summary, intestinal colonization affects the intestinal immune status in several ways: it leads to an overall decrease in

leukocyte presence, marked by a decrease of macrophages but a rise in neutrophil presence, and these changes of leukocyte populations are dependent on Myd88 mediated signaling. On a transcriptional level the effects of colonization are characterized by a suppression of *myd88* and downstream regulated genes in the intestine, but absence of pro-inflammatory *il1b* regulation (Supplementary Fig. 4b, d) .

**Host transcriptome responses to microbes is Myd88-dependent**. The central function of Myd88 in TLR signaling predicts that Myd88 plays an important role mediating the transcriptome alterations observed in the host upon microbial exposure. Considering that transcription of *myd88* itself is suppressed by conventionalization, we investigated its role in modulation the transcriptional response of the host to the microbiome. To pursue this question, we generated germ-free and conventionalized transcriptomics datasets from larvae in the *myd88* mutant background and compared these transcriptomics data sets to those of WT embryos of the zebrafish ABTL lineage. As this comparison is not between groups generated by crossing siblings, it cannot be ruled out that other parental genotype differences could contribute to the results. Therefore, using qPCR, we validated that the transcriptional regulation of *myd88* itself, in response to conventionalization, was retained in the WT siblings of *myd88* mutants analyzed in this transcriptomics data set (Supplementary Fig. 6). In the *myd88* mutant background the number of genes differentially regulated by microbial colonization was reduced from 257 to 84 (Supplementary Table 2). Interestingly, there was almost no overlap between the two groups; 78 of the 84 regulated transcripts in the *myd88* mutants could not be found regulated in the WT larvae. Thus, nearly the entire transcriptional response signature of the host to the microbiome seems dependent on Myd88 signaling (Supplementary Fig. 6).

In contrast to the transcriptome response to colonization in a WT background, there was a very clearly discernable ontology signature in the transcriptomic response of the *myd88* deficient larvae. Whereas the WT response included genes of several different functional classes, primarily transcription factors and intracellular signaling molecules, the 78 genes making up the *myd88* deficient response exhibited a very strong trend towards lipid metabolic processes (see Supplementary Table 7 and Supplementary Fig. 7 for comparative GO analysis). Most of these genes also differed in their expression levels between the WT and *myd88* mutants independently of microbial colonization status (Supplementary Tables 2 and 3). These data seem to indicate an important role for Myd88 in maintaining metabolic homeostasis, which is sensitive to colonization status. Considering our qPCR and in situ hybridization results, indicating that the gut and liver are important organs of *myd88* transcriptional upregulation under germ-free conditions (Fig. 2b, c, d), it may indicate an unappreciated role for Myd88 in mediating some of the intriguing effects of the microbiome on energy homeostasis.

**A common microbiota-responsive host transcriptome**. The germ-free and conventionalized larvae represent extremes on a scale of complexity of interactions between the host and the intestinal microbial communities, from an absence of microbial colonization and host-microbe interactions in germ-free larvae to the much greater complexity of uncharacterized communities in conventionalized larvae. To assess the effects of this complexity, we examined the impact of colonization by specific species on the transcriptomic response of the host. To this end, we compared the transcriptomic responses of germ-free larvae with two mono-associated groups of embryos, each colonized

with a single commensal bacterial species. The species were *Exiguobacterium* ZWU0009 (phylum Firmicutes) and *Chryseobacterium* ZOR0023 (phylum Bacteroidetes), both of which represent phylae of major research interest in humans and mice[33,34], though not numerically dominant species in zebrafish intestinal communities[8,23,35]. Thus, the mono-associated colonization groups are likely to represent very different microbial communities to those of the conventionalized group, and any transcriptional regulation that is shared between such diverse colonization conditions is likely to be very robust to differences in the nature of colonizing microbial communities. Successful enteric colonization in mono-associated embryos was verified microscopically using the amine-reactive Dye-light fluorescent labeling system (Fig. 3a). The transcriptomics results yielded 168 and 122 significantly regulated transcripts in response to mono-association with *Chryseobacterium* and *Exiguobacterium*, respectively, and show a very clear overlap with the host transcriptome response to conventionalization (Fig. 3b and Supplementary Tables 4 and 5). The magnitude of the responses in different conditions were more variable. Overlaying the sets of genes which met the regulation criteria of fold change and *P*-value yielded 65 genes shared among the three data-sets. All were regulated in the same direction, 60 down and 5 up (Fig. 3c, d). This shows that many of the transcriptional effects regulated by colonization by a complex microbial community can be mimicked by mono-association, an exposure that must be considered drastically different in nature from conventionalization. The identified set of 65 shared genes can be used as a common marker set of shared transcriptional response which seems independent of the nature or treatment dose of the stimulating microbiota.

**Suppression of Myd88 transcription is dependent on TLR2**. The sensitivity of *myd88* transcription to colonization points to the presence of a sensing mechanism whereby the host responds to the presence of microbes. While analyzing the shared transcriptional response marker set (Fig. 3c) we identified a large overlap with a set of 48 transcripts constituting the Tlr2 dependent response to the injection of the synthetic ligand Pam3CSK4 which mimics the bacterial lipopeptide[32] (see Fig. 4a). Out of the 65 shared response transcripts, 11 were shared with the Tlr2 dependent transcription response set. In contrast, there was no overlap with a Tlr5 dependent transcription response set of the same study[32]. To investigate whether Tlr2 could be involved in the microbial suppression of *myd88* transcription, we performed qPCR analysis to assess transcription of *myd88* and two other shared transcriptional response markers in conventionalized and germ-free larvae of Tlr2 loss of function mutant fish line (*tlr2*sa19423, see materials and methods) compared to wildtype siblings (Fig. 4c). The results show that the down regulation of *myd88* transcription in response to colonization only occurs in the WT siblings and not in Tlr2 mutants. In the Tlr2 mutant, the regulation of the two other selected markers of the shared transcriptional response, *cebpb* and *fosl1a*, was no longer significant. We performed extensive confocal laser scanning microscopy of isolated intestines of L-plastin stained Tlr2 mutant and control fish to verify that Tlr2 was involved in mediating the immune regulation in the intestine. These data show that the microbiome induced intestinal immune regulation, affecting intestinal leukocyte populations, was no longer apparent in Tlr2 mutants (Fig. 4d), confirming that Tlr2 and Myd88 are both implicated in the microbiome-mediated intestinal immune regulation. Taken together these data suggest the presence of a microbiome sensitive immune regulatory

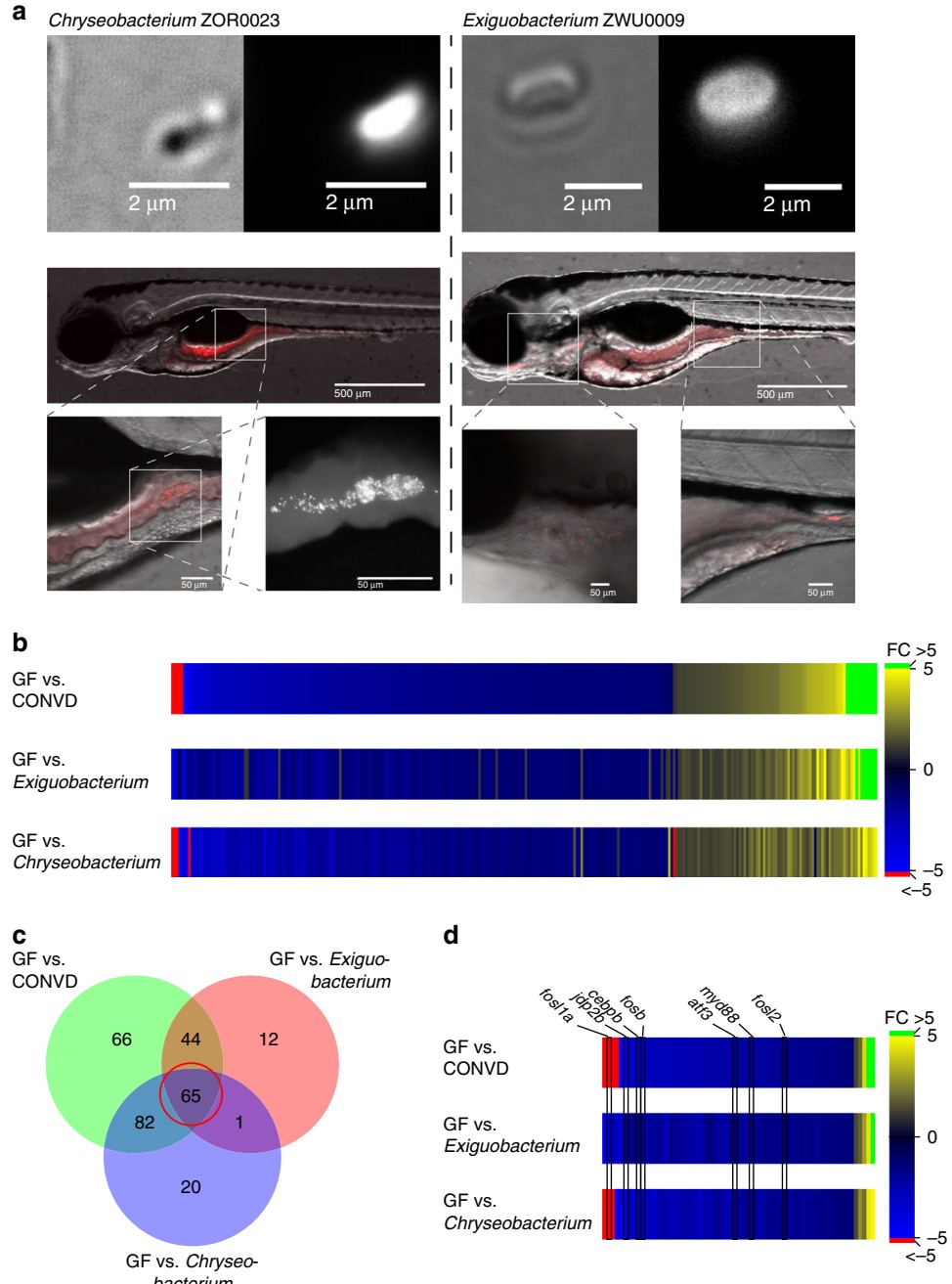

**Fig. 3** Mono-associated treatment groups help refine the colonization sensitive genes. **a** *Chryseobacterium* ZOR0023 (phylum Bacteriodetes) and *Exiguobacterium* ZWU0009 (phylum Firmicutes) visualized microscopically using the amine-reactive Dye-light fluorescent labeling system, colonizes different parts of the intestinal tract of the larvae. By plating homogenized embryos, the colonizing CFU burden was estimated to be approximately 70–75 for each, colonization patterns represent observations made in three independent colonization experiments. **b** Heat map displaying the normalized fold changes of 290 transcripts which reached statistical significance as differentially expressed in at least one of the colonized samples compared to germ-free. **c** Venn diagram showing the overlap in genes that make the significance cut-off for differential expression in the different colonized groups versus germ-free. **d** The central group of genes in the Venn diagram defines a group of 65 genes which all exhibit similar transcriptional responsiveness to bacterial colonization. These 65 genes represent strong candidates for markers of the shared transcriptional response. *Myd88*, along with several transcription factor encoding genes known to function downstream of Myd88, are among them

mechanism, functioning through Tlr2 and Myd88 to regulate intestinal leukocyte status in response to colonization.

## Discussion

Our results show that germ-free conditions are characterized by a lower number of neutrophils in the intestines compared to conventionalized conditions, in line with previous observations from similar zebrafish studies[9]. We show that the neutrophil responses of the intestine are dependent on Myd88. Additionally, we show a significant, Myd88 dependent, elevation of macrophage presence in the intestine under germ-free conditions.

The fact that the elevated intestinal macrophage infiltration in germ-free conditions is dependent on Myd88 is intriguing, as the

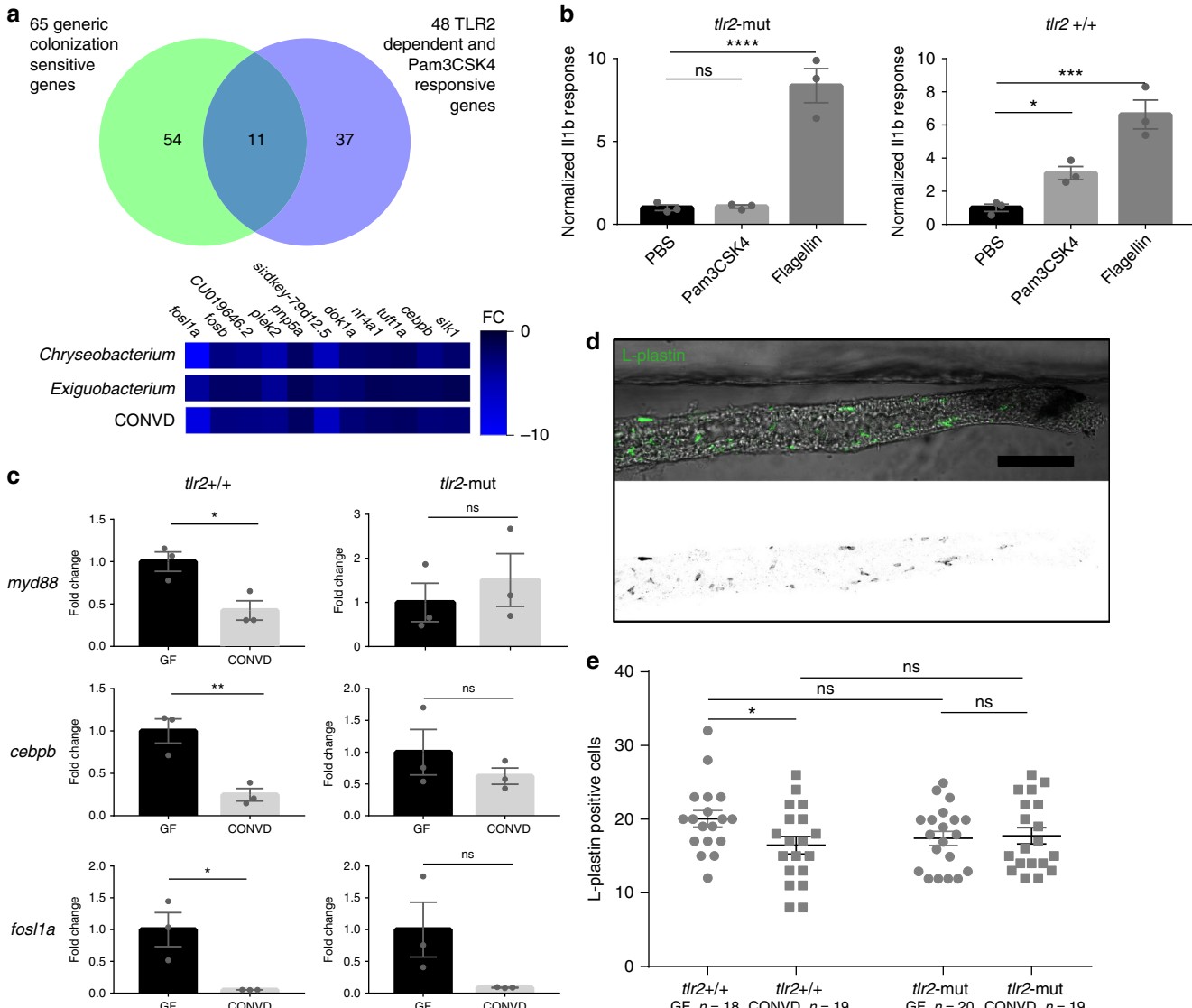

**Fig. 4** Microbiome transcriptional control of *myd88* is dependent on Tlr2. **a** Venn diagram showing the overlap between the identities of 65 primarily suppressed transcripts displaying transcriptional sensitivity to all the different colonization modes tested in this study, and thus are strong markers of the shared transcriptional response, with 48 primarily induced transcripts displaying Tlr2 dependent transcriptional sensitivity to injection of the synthetic ligand Pam3CSK4. 11 transcripts were shared among the two sets, all of which are suppressed by colonization, but induced by Pam3CSK4 in a Tlr2 dependent manner. **b** Q-PCR analysis comparing the inflammatory *il1b* response to injection of Tlr2 ligand (PAM3CSK4), Tlr4 ligand (LPS) and control (PBS) in Tlr2-mut and Tlr2 + / + 1 h after injection. (mean ± s.e.m., $n = 3$ biological replicates, 10 embryos per group), *$P \le 0.05$ ***$P \le 0.001$; ****$P \le 0.0001$, by Student's *t*-test. **c** Q-PCR analysis comparing the relative expression levels of *myd88, fosl1a* and *cebpb*, in conventionalized (CONVD) versus germ-free (GF) conditions in Tlr2 deficient mutants versus WT siblings. (mean ± s.e.m., $n = 3$ biological replicates, 15 embryos per group), *$P \le 0.05$; **$P \le 0.01$, by Student's *t*-test. **d** Representative image of intestines excised from L-plastin stained embryos. Scalebar represents 100 μm. **e** The elevated leukocyte presence under germfree conditions in the WT siblings was not observed in Tlr2 mutants. Figure is representative of three independent replications, *$P \le 0.05$; by two-way ANOVA with Bonferroni correction for multiple comparisons, error bars represent standard error of the mean

canonical function of Myd88 is to initiate inflammatory responses upon TLR stimulation, which is supposed to be absent under germ-free conditions. To resolve this conundrum, we analyzed the system wide transcriptomic responses mounted by the host in response to microbial colonization. Unexpectedly, we found that *myd88*, along with several other genes known to be regulated in a Myd88 dependent manner during bacterial infection, is among the down regulated genes in conventionalized larvae.

Having defined a gene marker set of the transcriptional response signature of the host to the microbiome and considering that suppressed *myd88* expression is a key element of this response, we investigated the function of Myd88 in the host response to microbial colonization. Results from *myd88* deficient mutants revealed that

Myd88 signaling is indispensable for this response. This dependence on Myd88 could reflect the simplicity of the larval colonization system we used, that is defined by never-fed conditions and thereby eliminating a number of confounding factors. While this experimental set-up allows us to draw strong conclusions regarding Myd88 dependence and innate immune cell responses, extrapolation of the results to more complex, conditions of microbiome interactions and colonization in the presence of an adaptive immune system and feeding should be done with caution. Nevertheless, the model is a good complement to present murine models as this model allows different factors to be added in sequentially. This can aid in dissecting the influences deriving from different factors such as specific microbial species, host genotypes or diet.

Using a gnotobiotic system we investigated whether the identified gene marker set for the transcriptional response signature of the host to the microbiome is dependent on microbiome diversity. The transcriptional response signature is remarkably similar, and, in most cases, the same genes are regulated in the same direction when comparing two mono-associations with conventionalized conditions. This is interesting given that the conventionalized communities, though not characterized, are assumed to be dominated by gamma proteobacteria which have been found to be numerically dominant in the intestines of young zebrafish larvae[36], while the Bacteroidetes and Firmicutes are much less prominent. In this way, we could define a generic set of 65 genes which display robust responsiveness to colonization by simple mono-association, as well as to complex undefined communities of the conventionalized groups (Fig. 3b, c). This generic set included *myd88* along with several of the other signature transcripts of shared transcriptional response (Supplementary Table 1 and Fig. 3d), indicating that microbiome suppression of *myd88* transcription, can also be exerted by both of the species of commensal bacteria tested in this study. Interestingly, a microarray transcriptome study of germ-free versus conventionalized conditions by Kanther et al.[10], a study in which the larval host had received feeding and analysis took place at 6 DPF, revealed little overlap of regulated transcripts with our shared response transcript set. Since our shared response gene set appears to be very robust to deviations in the composition of commensal microbial communities, it seems unlikely that all the differences are caused by the differences in microbial communities in the conventionalization protocol. Several other possible reasons might add to explaining these differences, including circadian regulation, zebrafish genetic strains or the introduction of feeding in the system.

In the *myd88* deficient background the transcriptomic response to microbial colonization exhibits a profile of expressional alterations with a striking enrichment in genes involved in lipid mobilization, bile production and reverse cholesterol transport (RCT) (Supplementary Table 3). This observation is very interesting considering the recent findings, that myd88 switches metabolic pathways toward obesity in response to nutritional status[21]. Our experiments suggest that microbial exposure and colonization of the intestine, in *myd88* deficient embryos strongly influences the transcriptomic levels of important genes involved in lipid and cholesterol metabolism, which are stable under the same treatment regime in WT embryos. These findings seem to indicate that Myd88, in addition to acting as an epithelial sensor of nutritional intake[21], also plays an important role in maintaining relatively steady transcriptional levels of genes involved in lipid metabolism, despite alterations in the composition of the enteric microbial communities. Interestingly many genes of these metabolic pathways affected by colonization in the *myd88* mutant were recently found to be affected by colonization in mutants of the *hnf4a* gene, which specifically binds and activates a microbiota suppressed intestinal epithelial transcriptional enhancer[37]. It will be interesting to study whether there is a functional link between myd88 mediated microbial recognition and the Hnf4 gene, and its link with inflammatory bowel diseases as shown by Davison et al.[37]. Though it is beyond the scope of this study to characterize in full the implications of *myd88* deficiency to the metabolic health of the host, we believe that it merits further experimental attention.

The canonical mechanism of TLR signaling upon ligand stimulation works by bringing the intracellular Toll/Interleukin-1 receptor (TIR) domains of the receptors into close proximity[38–40]. These domains of dimerized receptors serve as a fundamentally important scaffold on which adaptors, most importantly Myd88, can form the macromolecular protein complex known as the myddosome. In this complex Myd88 serves to relay the signal initiated by TLR dimer formation to the downstream signaling

partners of the interleukin-1 receptor-associated kinase (IRAK) family[41]. This occurs through homotypic protein domain interactions between the TIR domains of the ligand stimulated TLR dimers and Myd88, and between the DEATH domains of Myd88 and IRAK family members. It is a fundamental feature of this signaling mechanism that Myd88 does not spontaneously self-oligomerize and trigger ligand independent signaling[42–44]. In light of these established signaling events our results provide novel mechanistic insights that we have summarized in a model (Fig. 5). This model predicts the presence of a hitherto unknown negative feedback regulation mechanism, in which some pattern recognition receptor mediated signaling is responsible for the transcriptional down regulation of *myd88* in response to microbial associated molecular patterns (MAMPs), and the absence of such dampening MAMP signaling leads to the upregulation of *myd88*. Our results obtained in the Tlr2 mutant shows that this toll-like receptor is involved in the suppression of Myd88 expression exerted by the microbiome. This result gives support for the previously published function of TLR2 in establishing colonization by a commensal of the human microbiota[12]. Despite this realization the mechanism is not straightforward to comprehend. The 11 overlapping transcripts between the 65 shared transcriptional response markers and the 48 genes constituting the Tlr2 dependent response to PAM3CSK4 (Fig. 4), are universally oppositely regulated in response to conventionalization compared to the Tlr2 mediated response to the injected ligand, i.e., the *cebpb/d* and AP1 transcription factors are induced by PAM3CSK4[32] but suppressed by conventionalization. That means, in terms of Tlr2 dependent transcriptional regulation, that the germ-free state is more reminiscent of one where a PAMP has been injected. By extension, the introduction of microbes in this model suppresses a Tlr2-Myd88 driven immune status, which possibly causes the altered intestinal leukocyte infiltration. Several explanations could be proposed to account for this paradoxical observation that the absence of ligand in the germ-free state can lead to a transcriptional profile that resembles a ligand induced TLR2 response. It is conceivable that the signaling reflects a stimulation by an endogenous ligand. Myd88 is known to facilitate TLR mediated signaling of various damage-associated molecular patterns (DAMPs) in certain instances of wounding or cancer (see[45–47] for recent reviews). However, while DAMP signaling cannot be conclusively excluded, the absence of any obvious cause of tissue damage in the germ-free embryos, and the lack of evidence for the induction of inflammatory cytokines seems to argue against DAMP driven signaling as a cause, as they are generally considered proinflammatory in nature[45–47]. Rather, we propose that the elevated *myd88* could be the driver of a ligand independent response characterized by induction of the genes encoding the CCAAT/enhancer binding proteins and AP1 transcription factors. If this is the case, it would, to the best of our knowledge, be the first described in vivo example of this, though it has been shown in vitro[48]. Two mutually non-exclusive interpretations could be proposed: either Myd88 above a certain intracellular threshold concentration will self-oligomerize, independently of external signals, or an aberrant low-level myd-dosome formation might occur either at non-dimerized TLRs or at non-stimulated TLR dimers, generating a low base level of signaling, which is exacerbated by external stimulation, but can also be affected by elevated *myd88* transcription. Further studies are needed to elucidate the exact nature of such a feedback regulated mechanism, including the specific function of TLR2 in the intracellular relay mechanism.

## Methods

**Zebrafish maintenance and embryonic rearing**. Zebrafish were handled in compliance with animal welfare regulations and maintained according to standard protocols (http://zfin.org). The breeding of *myd88*−/− (*myd88*[hu3568]), and *tlr2*-

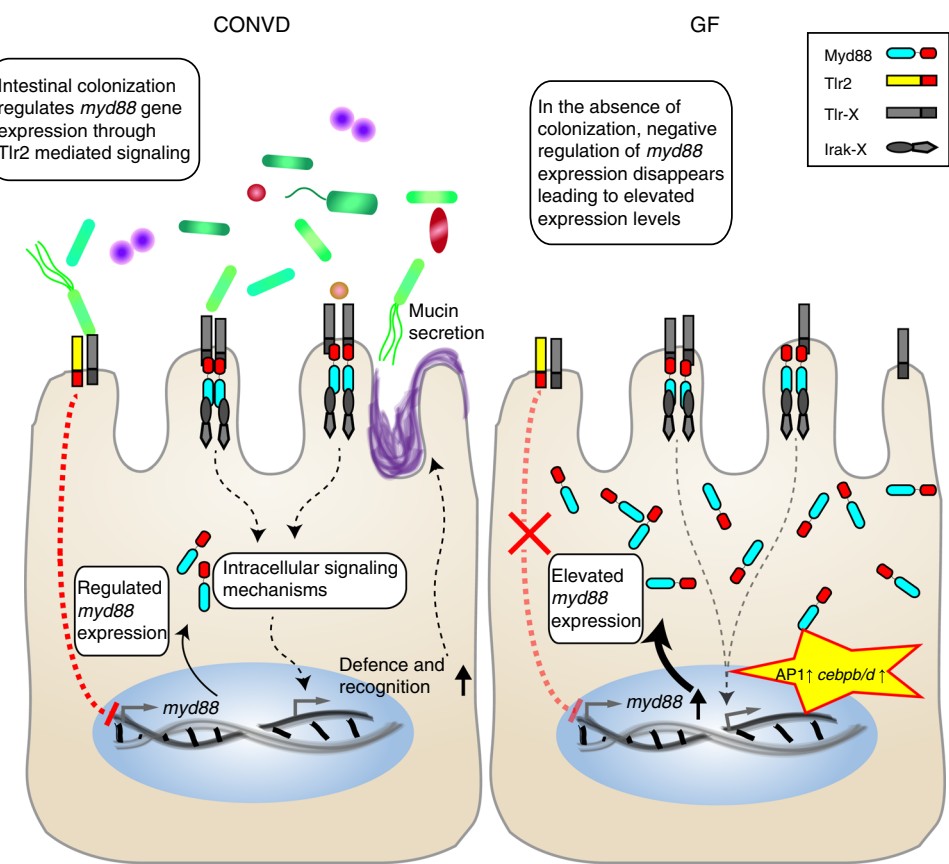

**Fig. 5** Model of colonization-driven Myd88-dependent transcriptional feedback mechanism. In the normal colonized state (CONVD), one or more microbial associated molecular pattern(s) is perceived by Tlr2 in an, as yet, unidentified receptor constellation. The resulting intracellular signaling events have a suppressing effect on *myd88* transcriptional activity, keeping the intracellular Myd88 concentration below what is permissible for it to induce ligand independent signaling events, yet still high enough for normal TLR stimulated responses to occur. Normal Tlr stimulated Myd88 dependent signaling leads to protective mucin secretion in the intestine. In the germ-free (GF) state, the absence of the Tlr2 mediated negative regulation of *myd88* transcription leads to elevated intracellular concentrations, sufficient to drive ligand independent signaling including transcripts associated with a Tlr2 dependent ligand stimulated response such as those encoding AP1 transcription factors and CCAAT/enhancer binding proteins

mutant (*tlr2^sa19423*) mutant fish was approved by the local animal welfare committee (DEC) of Leiden University (license number 10612, protocol 12232). *Myd88*−/− have been outcrossed at least three times to wildtype ABTL since entering our facility. In the *tlr2*-mutant fishline (*tlr2^sa19423*—https://zfin.org/ZDB-ALT-131217-14694) resulting from an ENU mutation screen from the Sanger Institute, a thymine to adenine point mutation, creates a premature stop codon at amino acid 549 in the C-terminus of the leucine-rich repeat (LRR) domain. The result is a truncated protein with no Toll/IL-1 receptor (TIR) domain, which cannot interact with Myd88 and Tirap (Mal)[49,50]. The mutant has been outcrossed three times to ABTL since entering our facility, the larvae in these experiments were the offspring of separated genotyped adults (−/− and +/+) from a hetero-zygous incross. The mutant is considered a loss of function mutant as it is found to phenocopy a morphant response[32] to TLR2 ligands (Fig. 4).

Germ-free and conventionalized embryo groups were generated essentially in accordance with the "Natural breeding method" previously described[24], with few deviations: (I) all embryonic groups were maintained at a density of three embryos per ml in autoclaved egg water (60 μg/ml Instant Ocean Sea Salt, (Spectrum Brands, Blacksburg, USA)). (II) to generate conventionalized groups, rather than introducing systems water directly, as described in ref.[24], we kept one dish of conventionally reared larvae (of WT (ABTL) parental cross which were reared in a non-sterile fashion) in sterilized egg water, with one daily water change, and maintained at the same density and temperature as treatment groups. At the time of conventionalization, egg water from this dish was used in a 1:50 dilution with sterilized egg water to conventionalize the larvae. This exposure corresponded to approximately $4 \times 10^3$ colony forming units (CFU) per ml. This CFU burden was estimated to correspond to approximately 50 CFU per embryo by plating of homogenized embryos. No media were changed from 3 to 5 DPF in any group. The assignment into groups was entirely random, with no consideration to larval behavior or appearance other than the exclusion of unhatched embryos and embryos with visible morphological deformities. Sterile conditions in germ-free groups were monitored by daily plating at least 2 ml swim water on tryptic soy agar and Luria-Bertani agar plates under aerobic conditions. Any corpses and shed

chorions were plated, and plates were incubated for 2 days at 28 °C. Processing of larval material for any purpose was performed at midday, approximately 4–6 h after first light exposure of the larvae. Only larvae appearing morphologically normal were included in any analysis.

**Generation of monoassociated larval treatment groups**. For monoassociated treatment groups specific bacteria (*Exiguobacterium* ZWU0009 and *Chryseo-bacterium* ZOR0023) were grown fresh over night at 28 °C on tryptic soy agar plates. Colonies were suspended in sterilized egg water and adjusted to a final $A_{600}$ optical density of 0.005, which corresponded to an approximate concentration of $2 \times 10^6$ and $12 \times 10^6$ CFU per ml for *Exiguobacterium* and *Chryseobacterium*, respectively. Larvae were kept in this water from 3 to 5 DPF. Monoassociation was tested by plating at day 5, by assessment of the uniform appearance of colonies on tryptic soy agar plates.

**Fluorescent staining of monoassociated colonizing bacteria**. For fluorescent labeling with Dylight 594 NHS Ester (Thermo Fisher, Waltham, USA), bacteria, suspended in sterilized eggwater (see above), were spun down and resuspended in $0.1$ M $Na_2CO_3$ (pH 8.3–9.0). Bacteria were subsequently spun down a second time, and incubated in 300 μl $0.1$ M $Na_2CO_3$ (pH 8.3–9.0) containing 6.5 μl of Dylight dye (10 mg/ml in DMSO) at room temperature for 2 h, shielded from light and with gentle agitation. After incubation with the dye, stained bacteria were washed twice with autoclaved egg water and resuspended at a final $A_{600}$ optical density of 0.005 which was used directly as colonization medium added to 3 DPF embryos. Staining had no apparent effect on bacterial viability or on the larvae. 2 days later, at 5 DPF, larvae were washed briefly in autoclaved egg water and kept in autoclaved egg water for an additional 2 h, after which they were imaged by confocal microscopy. Patterns have been validated independently in triplicate.

**Whole-mount in situ hybridization**. In situ hybridization was performed according to the published protocol by the Thisse group[51]. Each gene was tested by

two non-overlapping digoxigenin labeled anti-sense RNA probes, generated by in vitro transcription from T7 elongated PCR products Primer sequences (5′-3′): *myd88* probe #1: TCACGTACCTGGAGATCAAAAACTTCGAG/CCACTGGAACCTGAAGCGGTTTCCTC, *myd88* probe #2: GGACCTCGAACACAGGAGAGAGAAGG/CCAGGAAGGACGTCTCTGTCAAACACAC, *fosl1a* probe #1: GGCTCGAGCTCCGCGTCTGTCG/CGCAGCTGCTCTGATGACACCAGGC, *fosl1a* probe #2: TCTCTCCTGAGGAACTTGAGCGGCG/CTGAGTGATGGGATGTCATTGCTGGAGTCC, *jdp2b* probe #1: CGGTTTTCCCGCCACCACACTGAC/CTTTCCTTCTTTCTGTTTCGACAGCGAGC, *jdp2b* probe #2: GACAGACTTTCTGCAAAAGGAGTCCGAG/GGTGTCGACACAAATCCGTTTTCAACTTC, *cebpb* probe #1 TGCGTCCATGTCTGACATGTACAATCTGG/ACCGTTGACATGGACTCAATGTATGCGC, *cebpb* probe #2: CCAACACGTTTGCGCACAAGAGCGC/GTACTCCGGACTGTGCCTGTCCAC.

**L-plastin staining.** Polyclonal antibody against zebrafish L-plastin, produced by immunizing a rabbit as previously described[52], was a kind gift from Dr. Huttenlocher of the University of Wisconsin, and staining was performed on 5 DPF larvae fixed in 4% paraformaldehyde, using anti L-plastin primary antibody at 1:1000 dilution and Alexa 488 or 568 conjugated secondary antibody at 1:200 dilution as previously described[53]. Confocal stacks acquired at ×20 magnification formed the basis of the quantification of leukocytes in the distal intestine. Unless otherwise stated the area of counting was defined by counting 4 somites back from the cloaca.

**Total RNA isolation.** For whole organism transcriptomics and qPCR experiments 15 embryos were pooled together, snap frozen in liquid nitrogen and stored at −80 °C until extraction. For intestinal versus body qPCR analysis embryos intestines were extracted from anaesthetized 5 DPF embryos using sharpened forceps (see Supplementary Fig. 4A). Total RNA was extracted by trizol method (Invitrogen), according to the manufacturer's protocol. Each sample was further cleaned up on RNeasy mini spin column (Quiagen) according to the supplier's protocol. RNA integrity was assessed by bioanalyzer (Agilent); all samples were found to have a RIN ≥ 9.

**Illumina sequencing.** Total RNA isolated as described above was used to create RNAseq libraries. A total of 3 μg of RNA was used to make RNAseq libraries using the Illumina TruSeq RNA Sample Preparation Kit v2 (Illumina Inc., San Diego, USA). Two minor modifications were made to the protocol provided by the manufacturer. In the step describing the adapter ligation, 1 μl instead of 2.5 μl adaptor was used. In the library size selection step, the library fragments were isolated with a double Ampure XP purification with a 0.7 × beads to library ratio. The resulting RNAseq library was sequenced using an Illumina HiSeq2500 instrument in accordance with the descriptions provided by the manufacturer, with a read length of 1 × 50 nucleotides. Image analysis and base calling were done by the Illumina HCS version 1.15.1. Three biological replicates for each treatment regime were sequenced and mapped.

**Deep sequencing data mapping.** Illumina reads were analyzed using the Genetiles server[54]. In all cases more than 80% of the reads were successfully mapped to the ensemble GRCz10 ensembl zebrafish genome build (http://www.ensembl.org/Danio_rerio/Info/Index). Ratios of normalized read counts in germ-free versus conventionalized conditions were filtered applying a fold change cutoff of +/− 1.3 and a stringent significance cutoff of $P \leq 0.05$ after minimization of false discoveries based on the Benjamini Hochberg method (see Supplementary Tables 1, 2, 4, and 5).

**qPCR.** cDNA was generated from total RNA samples using the iScript cDNA synthesis kit (biorad). RNA samples were from independent experiments apart from the RNAseq samples.

Primers for *myd88* qPCR analysis were as follows (5′-3′): CAGTGGTGGACAGTTGTGGAC/GAAAGCATCAAAGGTCTCAGGTG. Measurements were normalized relative to the house-keeping control gene 18S ribosomal subunit: TCGCTAGTTGGCATCGTTTATG/CGGAGGTTCGAAGACGATCA which was found to be the best most stably performing out of 6 house-keeping primer pairs.

In the case of intestinal versus body tissue analysis measurements were normalized relative to two housekeeping genes; glyceraldehyde-3-phosphate dehydrogenase (*gapdh*) and ribosomal protein L13a (*rpl13a*):

*gapdh* (5′-3′): CGCTGGCATCTCCCTCAA/TCAGCAACACGATGGCTGTAG

*rpl13a* (5′-3′): TCTGGAGGACTGTAAGAGGTATGC/AGACGCACAATCTTGAGAGCAG

Each pair of housekeeping genes were identical to those designed and tested as previously published[55].

*fosl1a* (5′-3′): CTCAGCCCTCCCAATCACATCT/TACACTTCGCCGCAGCCATT

*cebpb* (5′-3′): GCAGGCAACCTATCACCTACATAC/CGCAAGTTTCACCGACTACAAGT.

Primers and assay conditions for *fosl1a* and *cebpb* qPCR analysis were identical to those previously published[32].

Primers for *il1b* qPCR analysis were as follows (5′-3′):

GAACAGAATGAAGCACATCAAACC/ACGGCACTGAATCCACCAC.

All qPCR detection was performed on a BioRad CFX96 machine following a two-step protocol with 40 cycles 95 °C melting temperature for 10 s and 58 °C annealing and amplification for 45 s. Results were analyzed using the ΔΔCt method.

**Toll-like receptor ligand injections.** Toll-like receptor ligands were injected into the circulation of embryos at 28 HPF by caudal vein injection. Embryos were anaesthetized in egg water containing 0.02% (w/v) buffered Tricaine (3-aminobenzoic acid ethyl ester; Sigma-Aldrich) and placed on 2% agarose in egg water. See ref.[56] for an instructive video of microinjection technique.

**Morpholino injections.** Antisense morpholino oligomer targeting pu.1[57] was obtained from Gene Tools. 1 nl of morpholino solution at 1 mM was injected into the yolk of zebrafish embryos at the 1–2 cell stage.

**Statistical analysis.** Deep sequencing data reads were analyzed using the Genetiles alignment and statistics package as previously described[54]. All P-values for deep sequencing data were adjusted to reduce the false discovery rate, by the Benjamini-Hochberg procedure. All further statistical analysis was carried out using GraphPad Prism 6.0 software (GraphPad Software, CA, USA). Data was analyzed for normal distribution by D'Agostino and Pearson normality test and F test to compare variances. For qPCR analysis, comparing two two-sided Student's t-tests were applied, in cases when significant differences in variance existed Welch's correction was applied. For intestinal leukocyte numbers group counts were analyzed by two-way ANOVA with Bonferroni correction for multiple comparisons. In the case of RNAseq results differences were considered significant only when the Benjamini-Hochberg adjusted P-value was lower than 0,05 and fold changes exceeded ±1.3. For all other statistical analyses significance was established as $P < 0.05$. *$P < 0.05$; **$P \leq 0.01$; ***$P \leq 0.001$; ****$P \leq 0.0001$.

## Data availability

The raw RNAseq data have been deposited in the NCBI GEO database with the accession number GSE82200.

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

## Acknowledgements

We thank John Rawls (Duke University, USA) for providing the strains of *Exiguobacterium* and *Chryseobacterium*, used for the commensal bacterium mono-association studies. We thank Annemarie Meijer (Leiden University, the Netherlands) for providing the *myd88* deficient fishline. We thank Anna Huttenlocher (University of Wisconsin, USA) for providing the L-Plastin antibody. We thank Daniel Rozen (Leiden University, the Netherlands) for reading the manuscript and providing useful textual feedback. This work was supported by the Danish National Research Foundation grant no. DNRF79.

## Author contributions

B.E.V.K., J.S., and H.P.S. devised experiments. B.E.V.K. conducted experiments and analyzed data. B.E.V.K. wrote the manuscript. J.S. and H.P.S. provided feedback and comments for the manuscript. S.Y. genotyped *tlr2* mutant fish. G.L. performed plastic embedding, sectioning, and microscopy of ISH stained larvae. All authors read and approved the manuscript.

## Additional information

**Competing interests:** The authors declare no competing interests.

