## [Peer Review File · Nature Communications]

Reviewers' comments:

Reviewer #1 (Remarks to the Author):

This manuscript seeks to advance our understanding of the roles of myd88 and tlr2 in host transcriptional and inflammatory responses to microbiota in zebrafish. Novel aspects of this manuscript include (1) discovery of elevated "macrophages" in the intestines of germ-free zebrafish which are normalized by microbiota colonization in a myd88- and tlr2-dependent manner; (2) a large set of new whole-animal RNA-seq datasets that define the impact of microbiota presence and composition, and myd88 and tlr2 genotype, on the whole-animal transcriptome; and (3) identification of a core set of zebrafish transcripts that are commonly regulated by the complex microbiota as well as two individual bacterial species. These findings advance the field of zebrafish host-microbiota interactions, and suggest interesting roles for myd88 and tlr2. However, reviewer enthusiasm for the manuscript was limited due to (1) the reliance on whole-animal RNA-seq (with minimal validation by WISH) which severely limits the mechanistic insights that can be drawn from the present study, and (2) the relatively minor advances afforded into myd88 and tlr2 function in host-microbiota interaction. Specific concerns are listed below.

Major concerns:

1. The greatest limitation of the present work is that it relies on whole-animal RNAseq data, which prevents the authors and reader from ascribing affected genes and processes to specific cells/tissues. The authors acknowledge this (Line 138), but they still go on to draw interpretations that assume that these pathways are affected in common cell types. For example, in Figure 2A and 5, the nature of the data presented here does not afford any definitive conclusions about pathways active in the intestinal epithelium nor any other individual cell. This type of summary cartoon is therefore inappropriate. The authors are encouraged to not include summary cartoons that infer shared localization of these or other affected pathways, unless extensive additional WISH or cell/tissue-specific qRT-PCR can be performed.
2. Linked to my major concern #1, the authors have not provided evidence that convincingly links the leukocyte data and the RNAseq data. It is interesting to see how myd88 and tlr2 genotype affect these phenotypes, but there is no real attempt to link the two phenotypes. How did neutrophil and macrophage transcripts behave in their RNAseq data? Are there other ways to link these two data types?
3. Figure 2B: The WISH data is of low quality - the staining shown looks similar to background staining that can be encountered in WISH. It would be much more convincing to show multiple animals per condition and probe, as well as negative control animals using sense probe. For these reasons, it was particularly alarming that the WISH methods section on Line 377 listed no negative control conditions such as sense probe controls. Without such controls, it's not appropriate to assign staining to an anti-sense probe. This is important, as these WISH images were the only data that attempted to refine tissues-of-origin for the reported transcript differences (see major concern 1)
4. It is insufficient to say that Lplatin+ mpx- cells are macrophages, as the L-platin+ population of cells in zebrafish have not been adequately characterized and may include cell types other than neutrophils and macrophages. Use of other macrophage markers/transgenics such as mpeg1 are required to support this claim. If such data is unavailable, then some other more general term should be used (not macrophages).
5. Line 338: It's not clear from this methods section if +/+ and -/- animals compared in RNA-seq or other analyses were (1) siblings from crosses of heterozygous parents, or (2) products of separate -/- x -/- or +/+ x +/+ crosses. This is important, as sibling controls (ie, 1) are strongly preferred, because genetic background between different parental pedigrees could be sufficient to underlie some of the phenotypes observed here. Based on the way the methods section is written, I suspect separate crosses were used, therefore pedigree effects could contribute to the observed differences between

+/+ and +/- animals (in both myd88 and tlr2 experiments).

6. The authors report interesting differences when comparing GF/CV animals under WT vs myd88 mutant conditions, and also WT vs tlr2 mutant. However, it is unclear if the mutant animals used in these experiments were generated at the same time as their respective wild-type controls. If not, other variables may be involved (e.g., seasonal changes in microbiota, genetic background and pedigree, etc.). Please clarify in Methods.

7. Line 217: What is the evidence that this tlr2 mutant allele is a null? No citation is given for this allele, suggesting this is the first characterization. Therefore more evidence needs to be provided to confirm that this allele is a strong loss of function or indeed a null. The Methods section cites a separate manuscript in preparation that includes a more detailed analysis of this mutant allele - however the authors may not call this a null allele without providing evidence or a citation.

8. Line 395: Negative control experiments for L-plastin antibody staining need to be described here too. Also, the antibody product number, batch number, and vendor must be reported.

9. Line 197 and Figure 3B: Clarify how many genes were found to be significantly differential in each of the two mono-associations - don't just selectively report the overlap. Indeed, these mono-association data should be listed similar to the conventionalized data in Supp Table 1.

10. Overall, the depth of analysis of their RNA-seq data is relatively small. Unbiased evaluation of GO terms and KEGG pathways are not provided, and full list of differentially expressed genes are only provided for GV vs CV wild-type, and NOT for myd88 mutants, or monoassociations. Also, it's not clear whether the gene list in Supp Table 2 is a complete list of genes differentially expressed in myd88 mutants, or a selective list generated by the authors. For any list of differentially expressed genes shown in a Venn diagram in the figures, a full list of those genes and their fold changes should be provided in the Supplementary Materials for the benefit of the reader and field.

Minor concerns:

- Line 376: I was expecting methods describing colonization of GF fish to produce the images shown in Figure 3A. Also, were any tests performed to confirm that the mono-associated animals remained mono-associated for the duration of the experiment (eg, CFU assessment)? Also, the CFU/animal (and media, if possible) should be reported for conventionalized animals. Finally, methods used to test sterility of GF fish cultures must be reported in Methods.

- Line 400 or elsewhere in Methods: The number of biological replicate samples per condition need to be described. Presumably more than one replicate was performed per condition, but this is not clear. Also, the data accession for the RNAseq data should be provided in Methods.

- Supp Fig 1: "gastric ventricles" is not a common term in zebrafish GI anatomy. Do you mean "intestinal bulb"? Best just to strike this phrase and just say "intestine".

- Figure 2 - zebrafish do not have stomachs.

- Line 150: The phrase "as suppression of an aberrant response of the host to germ-free conditions" seems to be an unnecessary generalization about a very complex multi-tissue and multi-cellular response. More nuanced language would be appropriate. On this topic, I found their use of the term "MCHT" to describe a discrete set of microbially-regulated genes to be unnecessary.

- Supp Tables 1 and 2: Please improve the legends for these tables to make it clear which direction the fold-changes are representing? That is, in these fold-changes, which condition is the numerator? This could be shown as a X/Y in the legend.

- Please follow zebrafish naming conventions for genes and proteins (eg, when referring to zebrafish tlr2, use italicized tlr2, not TLR2 as currently shown in Figure 4 and 5).

- Lines 227-230 and Figure 4D: It looks like L-plastin+ leukocyte numbers are highest in GF WT compared to the other three conditions, but no statistical test is provided to say whether GF WT are significantly higher than GF/CV tlr2 mutants. These tests should be performed and reported.

- Line 264: Bacteroidetes and Firmicutes should be capitalized.

- Line 279: Additional potential reasons for differences with the previous fed study could be

differences in microbiome composition, colonization method, media, zebrafish genetic strain and circadian time of tissue harvest.

- Line 296: Format references appropriately.
- Line 340: The allele number and reference of the *myd88* mutant should be provided.
- Line 319-322: Yet another possible explanation of these data is that these animals have viral infections (which may not be cured by the GF derivation process) which induce Tlr2/Myd88 dependent immune responses, but these anti-viral responses are overwhelmed by anti-bacterial/fungal responses that occur after colonization. Tlr2 is known to mediate host responses to viral infection, and viral loads were probably not measured by the authors (as this is very difficult to do). If this were so, other anti-viral immune response genes might be induced in GF, so the authors could look for these as potential evidence for or against this hypothesis. I recognize this is a difficult possibility to address, but the authors may wish to consider it.
- Line 350: Clarify if egg water was sterilized by autoclaving.
- The authors do not present a strong justification for the selection of the two individual bacterial strains chosen for RNA-seq analysis here. If they do not represent major species of the native microbiota, then why use them here?
- Line 399: How is "distal intestine" defined morphologically to ensure an equivalent region is assessed in each animal?
- Line 325: The authors suggest ligand-independent signaling in GF animals, but have they considered damage-associated molecular patterns (DAMPs) as potential ligands in GF animals?

Reviewer #2 (Remarks to the Author):

This article describes that the adaptor gene *myd88* can be involved in the microbiome control on the host transcriptome and macrophage infiltration of the zebrafish intestines.

I think that the results are very interesting and present a useful model to start to understand the complex interactions between immune responses and intestinal microbiome.

Several aspects should be addressed:

- The authors describe the decrease of leukocytes in intestine after microbial colonization and they conclude that these cells are macrophages, because they are not neutrophils. There are transgenic zebrafish lines to be sure about the identification of these cells. The authors could use them in order to better characterize this population.
- Which is the meaning of an increase of neutrophils and a decrease in macrophages after colonization?
- Could we speak about an intestinal inflammation in *Myd88* deficient fish with the transcriptomic response and the different number of immune cells?
- Why did the authors select the two bacterial strains for the monoassociated larval treatments?
- Do *myd88* deficient fish show transcriptomic differences compared with WT, both in GF conditions and after colonization? It is not clear in the manuscript. For example, in Supp. Fig 2, are these genes all the differentially expressed after colonization in *myd88*^{-/-}? Are genes related with lipid metabolism differentially expressed between WT and *myd88*^{-/-} fish?

We are very grateful to the reviewers for their very helpful feedback. In the following we answer the comments of reviewers #1 and #2 in turn, the text in blue is our answers.

As the token access code has been removed from the revised manuscript we have added it here so that the reviewers will be able to access the raw data should they so wish. GEO database accession number: GSE82200, token reviewer access code: idsvuukwdxyzbcf

This document has its own reference list at the end. Please consult this list for references even in the cases when references appear in quotes from the manuscript text.

Reviewers' comments:

Reviewer #1 (Remarks to the Author):

This manuscript seeks to advance our understanding of the roles of *myd88* and *tlr2* in host transcriptional and inflammatory responses to microbiota in zebrafish. Novel aspects of this manuscript include (1) discovery of elevated "macrophages" in the intestines of germ-free zebrafish which are normalized by microbiota colonization in a *myd88*- and *tlr2*-dependent manner; (2) a large set of new whole-animal RNA-seq datasets that define the impact of microbiota presence and composition, and *myd88* and *tlr2* genotype, on the whole-animal transcriptome; and (3) identification of a core set of zebrafish transcripts that are commonly regulated by the complex microbiota as well as two individual bacterial species. These findings advance the field of zebrafish host-microbiota interactions, and suggest interesting roles for *myd88* and *tlr2*. However, reviewer enthusiasm for the manuscript was limited due to (1) the reliance on whole-animal RNA-seq (with minimal validation by WISH) which severely limits the mechanistic insights that can be drawn from the present study, and (2) the relatively minor advances afforded into *myd88* and *tlr2* function in host-microbiota interaction. Specific concerns are listed below.

Major concerns:

1. The greatest limitation of the present work is that it relies on whole-animal RNAseq data, which prevents the authors and reader from ascribing affected genes and processes to specific cells/tissues. The authors acknowledge this (Line 138), but they still go on to draw interpretations that assume that these pathways are affected in common cell types. For example, in Figure 2A and 5, the nature of the data presented here does not afford any definitive conclusions about pathways active in the intestinal epithelium nor any other individual cell. This type of summary cartoon is therefore inappropriate. The authors are encouraged to not include summary cartoons that infer shared localization of these or other affected pathways, unless extensive additional WISH or cell/tissue-specific qRT-PCR can be performed.

We thank the reviewer for this comment, and for the suggestions of strategies to address it. As this is indeed an important concern we have taken several approaches to validate that the representations in our graphics are accurate, at least for certain representative transcripts:

We have added transverse sections of ISH results for *myd88* to demonstrate the intestinal expression of the transcript. These images have been added to figure 2, and quite clearly reveal that the ISH signal is located to the intestine.

To further address the subject of tissue specificity we have performed qPCR analysis of *myd88*, as well as *Fos1a*, *Cebpb* and *Il1b*, performed on excised intestinal tissues (GI tract, liver, pancreas) alongside with samples from their respective remaining body tissues. *Fos1a*, *Cebpb* and *Il1b* were chosen as representative transcripts for the AP1 transcription complex, CCAAT/enhancer binding protein (C/EBP) family and the proinflammatory cytokines respectively, which are mentioned in the cartoon (2A). The *Myd88* result has been added to the revised figure 2, while the other qPCR results have been added to supplementary figure 4. The qPCR results confirm the significant intestinal regulation, of *Myd88*, *fosl1a* and *Cebpb* as compared to the remaining body tissues. Also, in keeping with the message of the manuscript and figure 2, *Il1b* is not significantly regulated, indicating that this phenomenon is not an overtly inflammatory one.

We recognize that this still does not indisputably demonstrate that the transcriptional regulation takes place in the same cells, as we have only demonstrated this at the level of tissues and only for representative transcripts. We have added the following cautionary sentence to the figure label of figure 2: “It should be noted that the transcriptional foldchanges are derived from whole-body transcriptomics and that it cannot be concluded that the transcriptional changes represented here all take place in the same cells, even though they are part of the same regulatory pathway”. As the figure label clearly states this caution, and the cartoon in figure 2 only purports to represent the canonical *Myd88* dependent TLR signaling pathway, we believe that the data-support is sufficient to merit keeping the cartoon, as it is important to the description of the regulation.

Regarding figure 5 we believe that the figure heading already provides sufficient caution by describing the figure as a “Proposed model”, which by its nature is somewhat speculative. With the demonstration that *Myd88* and representatives of the major downstream classes of transcripts that the manuscript deals with are intestinally regulated, we believe that it is justified to use such a graphical tool to facilitate the discussion.

2. Linked to my major concern #1, the authors have not provided evidence that convincingly links the leukocyte data and the RNAseq data. It is interesting to see how *myd88* and *tlr2* genotype affect these phenotypes, but there is no real attempt to link the two phenotypes. How did neutrophil and macrophage transcripts behave in their RNAseq data? Are there other ways to link these two data types?

This is a good point. As the altered intestinal leukocyte populations resulting from colonization as presented in figure 1, is shown to be *Myd88* dependent, we think the connection is clearer in this revised version. The addition of tissue specific qPCR results and sectioned ISH results for *myd88* (figure 2 C and D) very clearly demonstrates the regulation, in the intestine, of *myd88*. With the verification that the phenotype is not driven by *Il1b* (supplementary figure 4), we can arrive at a description of the phenomenon as a *Myd88* dependent response to colonization which is characterized by the increase in neutrophils and the decrease of macrophages, which is not overtly inflammatory in nature.

We recognize that though this description does provide a link between the leukocyte observations and the transcriptional data, it does not explain the shift in leukocyte population. Considering neutrophil and macrophage transcripts more broadly is an interesting approach and we are grateful to the reviewer for the suggestion.

To try to assess macrophage and neutrophil transcripts in the broadest possible sense, we utilized the annotation source of ZFIN by retrieving all transcripts that carried either the “macrophage” or

“neutrophil” descriptors. This approach returned 257 and 199 transcripts for macrophages and neutrophils respectively, with 61 transcripts figuring in both. Comparing these lists to those identified as significantly regulated in response to conventionalization in WT and *Myd88* deficient mutants (supplementary tables 1 and 2 respectively), we identified all the macrophage and neutrophil associated transcripts that were regulated in these analyses. This revealed that relatively few transcripts with known functions in macrophages or neutrophils were regulated in response to colonization. The identities of macrophage and neutrophil transcripts that were regulated in response to colonization have been compiled in a supplementary table (supplementary table 8).

Published work¹ has identified transcripts upregulated in classically (M1) and alternatively activated (M2) macrophages in zebrafish. None of those transcripts figure among the regulated transcripts to provide a possible explanation for the observed phenotype. So overall, we do not believe that the intestinal leukocyte phenotypes observed are linked through anything that can be described as an inflammatory status due to the absence of an *Il1b* response, or a polarized activation state. Rather we suggest it reflects an intestinal tissue regulatory phenomenon linked to systemic responses. Among the few macrophage and neutrophil transcripts found to be regulated in response to colonization is immunoresponsive gene 1 like (*Irg1l*) which has been described to serve a function in metabolic regulation of immune cell migration in epidermal cells². This could be an interesting candidate gene to focus on in further investigations of this phenomenon, but to truly elucidate the connection, it may be necessary to focus more specifically on the transcriptional regulation of the intestine and intestinal leukocytes. With new single cell in situ technologies being currently developed in other labs, it will in the future be possible to probe deeper into the status of intestinal leukocytes, and our results may serve as a starting point to further investigations of Tlr2-Myd88 signaling at the cellular level.

3. Figure 2B: The WISH data is of low quality - the staining shown looks similar to background staining that can be encountered in WISH. It would be much more convincing to show multiple animals per condition and probe, as well as negative control animals using sense probe. For these reasons, it was particularly alarming that the WISH methods section on Line 377 listed no negative control conditions such as sense probe controls. Without such controls, it's not appropriate to assign staining to an anti-sense probe. This is important, as these WISH images were the only data that attempted to refine tissues-of-origin for the reported transcript differences (see major concern 1)

Our WISH control was based on the observation of identical, spatially restricted patterns of expression resulting from the application of two non-overlapping antisense probes for each transcript. This is our preferred method of control in WISH experiments, as sense-probes may provide false positive results due to natural antisense transcripts (see Sun et al.³ for a recent review). This method of control is also the recommendation provided in the highly cited Thisse protocol that we have followed in these experiments⁴. We have added more extensive documentation of our WISH efforts in supplementary figure 3.

Furthermore, the added clarity of the 2 μ m plastic embedded sections of ISH results for *Myd88* (figure 2C) which very clearly demonstrates the specific pattern of intestinal and liver expression as well as the visible difference in signaling intensity between germ-free and conventionalized samples, supports our claim that this is a tissue specific expression and not unspecific background staining.

In further support of the validity of our reported WISH patterns i.e. expression in the liver and intestines at 5 DPF, we would like to point the reviewers' attention to the following peer-reviewed WISH results of other research groups:

Chris Hall and coworkers⁵ reported *Myd88* expression in liver and intestine of 4 DPF larvae by WISH.

Li et al 2010⁶ reported *Cebpb* expressed in liver and intestine. Furthermore, large directly uploaded expression datasets from Rauch et al 2003, and Thisse et al 2001, available at Zfin, have reported identical findings for *Cebpb* at 5 DPF.

4. It is insufficient to say that Lplastin+ mpx- cells are macrophages, as the L-plastin+ population of cells in zebrafish have not been adequately characterized and may include cell types other than neutrophils and macrophages. Use of other macrophage markers/transgenics such as *mpeg1* are required to support this claim. If such data is unavailable, then some other more general term should be used (not macrophages).

We are grateful for the comment and for the suggestion of alternative analysis method. We have followed this suggestion and carried out a similar analysis to that in figure 1, only utilizing the TG(*Mpeg1:mCherryF/Mpx:GFP*) double fluorescent line to simultaneously label macrophages and neutrophils in the intestines of germ-free and conventionalized embryos. The result, which has been added as supplementary figure 1 D, clearly demonstrates that the leukocyte population in the intestine shifts with fewer macrophages and more neutrophils in response to colonization.

5. Line 338: It's not clear from this methods section if +/+ and -/- animals compared in RNA-seq or other analyses were (1) siblings from crosses of heterozygous parents, or (2) products of separate -/- x -/- or +/+ x +/+ crosses. This is important, as sibling controls (ie, 1) are strongly preferred, because genetic background between different parental pedigrees could be sufficient to underlie some of the phenotypes observed here. Based on the way the methods section is written, I suspect separate crosses were used, therefore pedigree effects could contribute to the observed differences between +/+ and -/- animals (in both *myd88* and *tlr2* experiments).

This is a good point. To be clear: The *myd88* mutants in which the *myd88*-/- transcriptomics data-set were generated were not siblings of the (ABTL) wildtypes that form the basis of the other transcriptomics analyses. The line has been outcrossed to the ABTL background several times, but it is true that the parental pedigree might have effects on the RNAseq analysis. All other experiments were conducted comparing *myd88*+/+ to -/-. ie. Control number 1.

In the revised version, we have performed qPCR validation to ensure that the transcriptional regulation of *Myd88* in response to conventionalization is not affected by pedigree effects between the ABTL line used in WT experiments and the WT siblings of the *myd88* mutants. The result has been added to the supplementary figure 6 which deals with the transcriptomic analysis of *myd88* mutants. This is supporting the major claim of the paper on *myd88* regulation.

To clarify in the text, we have added the following sentence in lines 166-171:

“As this comparison is not between groups generated by crossing siblings, it cannot be ruled out that other parental genotype differences could contribute to the results. Therefore, using qPCR, we

validated that the transcriptional regulation of *myd88* itself, in response to conventionalization, was retained in the WT siblings of *myd88* mutants analyzed in this transcriptomics data set”

In the interpretation of the RNAseq we have made the statement regarding all the other regulated transcripts, that we have not directly confirmed, less strong by modifying the following sentence at lines 175-176:

“Thus, nearly the entire MCHT signature seems dependent on Myd88 signaling (supplementary figure 6).” The modification being that we have substituted the word “is” for “seems”.

As for the TLR2 mutant we have clarified in the methods section, lines 369-371 with the following text:

“The mutant has been outcrossed three times to ABTL since entering into our facility, the larvae in these experiments were the offspring of separated genotyped adults (-/- and +/+) from a heterozygous incross.”

6. The authors report interesting differences when comparing GF/CV animals under WT vs *myd88* mutant conditions, and also WT vs *tlr2* mutant. However, it is unclear if the mutant animals used in these experiments were generated at the same time as their respective wild-type controls. If not, other variables may be involved (e.g., seasonal changes in microbiota, genetic background and pedigree, etc.). Please clarify in Methods.

This is an interesting point. We do not think that seasonal changes to the microbiota is a likely cause of our observations for several reasons: 1) The transcriptional regulation is very robust, even to very significant differences in the nature of colonization. The regulation is essentially retained (i.e. the same transcripts are regulated, and the direction of regulation is nearly completely retained) between the CONVD, *Exigubacterium* and *Chryseobacterium*. These are fundamentally different conditions of colonization both in terms of the complexity of the composition of communities and in terms of absolute numbers of microbes. Therefore, we do not believe that seasonal variation that could be associated with conventionalization at different times of the year would be a significant factor. 2) The different results achieved in the mutant analyses, be they leukocyte-focused or transcriptional, are in each case temporally matched with their WT controls. The one exception to this is the RNAseq transcriptional analysis of the *myd88* mutant response to conventionalization, which we acknowledge did take place approximately one month after the WT transcriptomics did. However, qPCR validation confirming our major conclusion on *myd88* has been performed as completely independent experiments, showing that seasonal effects are not causing our described regulation. For other genes from the RNAseq data-sets the conclusion has been modified as described above.

We have added the following sentence to the materials and methods section under qPCR line 471:

“RNA samples were from independent experiments apart from the RNAseq samples.”

The analysis regarding leukocyte behavior in TLR2 -/- versus +/+ (figure 4 D), was carried out simultaneously in both groups. These larvae resulted from incrossing +/+ and -/- siblings of a heterozygous incross. The material for the qPCR analysis of TLR2 mutants in figure 4 C, was generated at the same time and in the same manner.

7. Line 217: What is the evidence that this *tlr2* mutant allele is a null? No citation is given for this allele, suggesting this is the first characterization. Therefore more evidence needs to be provided to confirm that this allele is a strong loss of function or indeed a null. The Methods section cites a separate manuscript in preparation that includes a more detailed analysis of this mutant allele - however the authors may not call this a null allele without providing evidence or a citation.

We appreciate this concern as this paper gives the first characterization of the mutant. Therefore, we have added a verification experiment comparing the response, in mutants versus their WT siblings, to the injection of the well described TLR2 ligand PAM3CSK4, as measured by qPCR assessment of IL1B transcription. The experiment uses PBS injection as a negative control, and the TLR4 ligand LPS as a positive control. The findings clearly demonstrate that the TLR2 mutant is indeed a null mutant, as the response to PAM3CSK4 is absent, while the inflammatory response to LPS is unaltered. The result of this experiment has been added to figure 4 of the main text, which deals with the TLR2 mutant. Furthermore, we have added more descriptive text to the Materials and Methods section in lines 364-369:

“In the *tlr2*^{-/-} mutant fishline (*tlr2*^{sa19423} - <https://zfin.org/ZDB-ALT-131217-14694>) resulting from an ENU mutation screen from the Sanger Institute, constituting a thymine to adenine point mutation, creating a premature stop codon at amino acid 549 in the C-terminus of the leucine-rich repeat (LRR) domain. The result is a truncated protein with no Toll/IL-1 receptor (TIR) domain, which cannot interact with Myd88 and Tirap (Mal)^{7,8}. The mutant is found to phenocopy a morphant response⁹ to TLR2 ligands (figure 4).”

With this addition we no longer to refer to the other manuscript in preparation, that is only concerned with infectious phenotypes of TLR2 mutants.

8. Line 395: Negative control experiments for L-plastin antibody staining need to be described here too. Also, the antibody product number, batch number, and vendor must be reported.

The antibody was a kind gift from Professor Anna Hüttenlocher at the university of Wisconsin, and does not have product number, batch number or vendor. We have added the following description to the materials and methods section lines 432-433:

“Polyclonal antibody against zebrafish L-plastin, produced by immunizing a rabbit as previously described¹⁰, was a kind gift from Dr. Huttenlocher of the University of Wisconsin”

We have added two control experiments to demonstrate the specificity of the antibody as supplementary figure 2. The controls compare the result of a staining carried out without primary antibody, and the effect of carrying out the staining in embryos that has been rendered immunodeficient by injection of a morpholino targeting the transcription factor Pu.1, which causes a blockage of myelopoiesis¹¹. The results demonstrate that the staining protocol is free of unspecific staining and the near complete depletion detectable signal in *pu.1* morpholino injected embryos demonstrates the specificity of the antibody.

9. Line 197 and Figure 3B: Clarify how many genes were found to be significantly differential in each of the two mono-associations - don't just selectively report the overlap. Indeed, these mono-association data should be listed similar to the conventionalized data in Supp Table 1.

We have added the numbers of significantly regulated transcripts in the text and added further supplemental tables listing the genes found to be significantly regulated in any analysis mentioned. The tables for mono-associations have been added as supplementary tables 4 and 5.

10. Overall, the depth of analysis of their RNA-seq data is relatively small. Unbiased evaluation of GO terms and KEGG pathways are not provided, and full list of differentially expressed genes are only provided for GV vs CV wild-type, and NOT for myd88 mutants, or monoassociations. Also, its not clear whether the gene list in Supp Table 2 is a complete list of genes differentially expressed in myd88 mutants, or a selective list generated by the authors. For any list of differentially expressed genes shown in a Venn diagram in the figures, a full list of those genes and their fold changes should be provided in the Supplementary Materials for the benefit of the reader and field.

We have added a supplementary table (supplementary table 7) of gene ontology terms which are enriched in each of the data sets mentioned. Furthermore, we have added a supplemental figure which provides a graphical representation of the overlap of GO Terms among the different colonization data sets in the form of a Venn diagram.

Minor concerns:

- Line 376: I was expecting methods describing colonization of GF fish to produce the images shown in Figure 3A. Also, were any tests performed to confirm that the mono-associated animals remained mono-associated for the duration of the experiment (eg, CFU assessment)? Also, the CFU/animal (and media, if possible) should be reported for conventionalized animals. Finally, methods used to test sterility of GF fish cultures must be reported in Methods.

Description has been added to the Materials and Methods section under “Fluorescent staining of monoassociated colonizing bacteria”, lines 406-408.

Mono-association was confirmed by assessing the uniformity of colony appearance on plates, and CFU assessment was performed at the end of the experiment. Details have been added to the Materials and Methods section “Generation of monoassociated larval treatment groups”, lines 399-400.

Conventionalization was assessed to amount to approximately 50 CFU, after homogenization of embryos and plating on TSA plates. And sterility in germ-free populations was monitored daily by plating egg water, shed corions and corpses on LB and TSA plates. Descriptions have been added to the Materials and Methods section “Zebrafish maintenance and embryonic rearing”, lines 387-390.

- Line 400 or elsewhere in Methods: The number of biological replicate samples per condition need to be described. Presumably more than one replicate was performed per condition, but this is not clear. Also, the data accession for the RNAseq data should be provided in Methods.

Done.

- Supp Fig 1: "gastric ventricles" is not a common term in zebrafish GI anatomy. Do you mean "intestinal bulb"? Best just to strike this phrase and just say "intestine".

The label has been changed to intestine.

- Figure 2 - zebrafish do not have stomachs.

The word has been changed to intestine.

- Line 150: The phrase "as suppression of an aberrant response of the host to germ-free conditions" seems to be an unnecessary generalization about a very complex multi-tissue and multi-cellular response. More nuanced language would be appropriate. On this topic, I found their use of the term "MCHT" to describe a discrete set of microbially-regulated genes to be unnecessary.

We certainly do not want to oversimplify a complex multicellular response, we are just attempting to provide a comprehensive interpretation of a complex RNAseq dataset. To make our intentions clearer we have modified this sentence, lines 156-158:

“This microbiome control of the host transcriptome (from here on abbreviated as MCHT) could be considered as suppression of an aberrant response of the host to germ-free conditions.”

Similarly the abbreviation “MCHT” is only meant to make the paper more readable, and we are open to suggestions for other descriptive abbreviations that describes the transcriptome signature.

- Supp Tables 1 and 2: Please improve the legends for these tables to make it clear which direction the fold-changes are representing? That is, in these fold-changes, which condition is the numerator? This could be shown as a X/Y in the legend.

Done

- Please follow zebrafish naming conventions for genes and proteins (eg, when referring to zebrafish *tlr2*, use italicized *tlr2*, not TLR2 as currently shown in Figure 4 and 5).

Done.

- Lines 227-230 and Figure 4D: It looks like L-plastin+ leukocyte numbers are highest in GF WT compared to the other three conditions, but no statistical test is provided to say whether GF WT are significantly higher than GF/CV *tlr2* mutants. These tests should be performed and reported.

Done

- Line 264: Bacteroidetes and Firmicutes should be capitalized.

done

- Line 279: Additional potential reasons for differences with the previous fed study could be differences in microbiome composition, colonization method, media, zebrafish genetic strain and circadian time of tissue harvest.

Good point.

- Line 296: Format references appropriately.

done

- Line 340: The allele number and reference of the *myd88* mutant should be provided.

Done

- Line 319-322: Yet another possible explanation of these data is that these animals have viral infections (which may not be cured by the GF derivation process) which induce Tlr2/Myd88 dependent immune responses, but these anti-viral responses are overwhelmed by anti-bacterial/fungal responses that occur after colonization. Tlr2 is known to mediate host responses to viral infection, and viral loads were probably not measured by the authors (as this is very difficult to do). If this were so, other anti-viral immune response genes might be induced in GF, so the authors could look for these as potential evidence for or against this hypothesis. I recognize this is a difficult possibility to address, but the authors may wish to consider it.

A very interesting and insightful comment. We have no indication that our fish carry a viral infection. The expression of the well described virus specific interferons of the interferon phi family (*ifnphi1*, *ifnphi2*, *ifnphi3* and *ifnphi4*) were all uniformly and very lowly expressed across all treatment groups (0-25 transcripts mapped for each of the 4 genes). Considering that each sample was generated from 15 whole embryos, this amounts to an exceedingly low level of transcription, which we take as an indication that a viral infection in our facility is very unlikely.

- Line 350: Clarify if egg water was sterilized by autoclaving.

It was. The clarification has been added.

- The authors do not present a strong justification for the selection of the two individual bacterial strains chosen for RNA-seq analysis here. If they do not represent major species of the native microbiota, then why use them here?

Our reasoning for picking the *Exiguobacterium* and *Chryseobacterium* strains was twofold:

1) We wanted our transcriptomics data sets to be useful to the microbiome community in a broad context. These strains represent the phyla Firmicutes and Bacteroidetes which are the focus of significant research interest in the microbiome field. Upon publication these data sets could allow comparative research into the unifying features of these phyla in the context of host-microbe interactions across the various model organisms and study designs and might thus serve as a useful resource to the field at large.

2) The fact that these species are not numerically dominant in the natural situation is important to the interpretation of the comparative analysis of the transcriptomics data, because we were aiming to analyze which part of the host transcriptomic response to colonization is conserved across very different colonization conditions. Therefore, it becomes important that the mono-association data sets are both fundamentally different from each other and from the conventionalized group. That we may confidently conclude that the 65 transcripts that we found significantly regulated in all these three data sets, constitute a conserved reflection of the host response to colonization in the broadest terms.

We see this as a strength of this analysis, even though it obviously has its limitations: Another study design might have focused more on mono-association conditions that more closely resemble the

composition of conventionalization conditions. We also recognize that distant from conventionalization conditions though our mono-associations may be, they are still considered commensal, and we do not know how the transcriptomic profile might differ if the mono-association had been with a pathogenic strain. As follow up studies such study designs may add further clarity. Ultimately though, any study design has its strengths and limitations. As we see it, the choice of mono-association groups allows us to define a restricted set of transcripts that robustly respond to very diverse colonization and has the potential to inform the wider field in the process of dissecting and characterizing the nature of the very complex host-microbe interactions.

We have added the following text to the manuscript to try to make these deliberations appear clearer in lines 206-210:

“Thus, the mono-associated colonization groups are likely to represent very different microbial communities to those of the conventionalized group, and any transcriptional regulation that is shared between such diverse colonization conditions is likely to be very robust to differences in the nature of colonizing microbial communities.”

- Line 399: How is "distal intestine" defined morphologically to ensure an equivalent region is assessed in each animal?

It was defined by counting 4 somites back from the colaca. Description has been added.

- Line 325: The authors suggest ligand-independent signaling in GF animals, but have they considered damage-associated molecular patterns (DAMPs) as potential ligands in GF animals?

An interesting point, and indeed one that we cannot definitively exclude. We do not, however, believe the responses that we observe are caused by an endogenous ligand, because this would in most described cases lead to an inflammatory state. While we have thoroughly refuted the possibility of an *il1b* driven inflammatory state, we recognize that we have not exhaustively ruled out the possibility of an inflammatory state driven by any other proinflammatory cytokine. Recognizing this point, we have added the following text to the discussion, lines 336-345:

“Several explanations could be proposed to account for this paradoxical observation; that the absence of ligand in the germ-free state can lead to a transcriptional profile that resembles a ligand induced TLR2 response. It is conceivable that the signaling reflects a stimulation by an endogenous ligand. Myd88 is known to facilitate TLR mediated signaling of various damage associated molecular patterns (DAMPs) in certain instances of wounding or cancer (see ¹²⁻¹⁴ for recent reviews). However, while DAMP signaling cannot be conclusively excluded, the absence of any obvious cause of tissue damage in the germ-free embryos, and the lack of evidence for the induction of inflammatory cytokines seems to argue against DAMP driven signaling as a cause, as they are generally considered proinflammatory in nature¹²⁻¹⁴.”

Reviewer #2 (Remarks to the Author):

This article describes that the adaptor gene *myd88* can be involved in the microbiome control on

the host transcriptome and macrophage infiltration of the zebrafish intestines.

I think that the results are very interesting and present a useful model to start to understand the complex interactions between immune responses and intestinal microbiome.

Several aspects should be addressed:

- The authors describe the decrease of leukocytes in intestine after microbial colonization and they conclude that these cells are macrophages, because they are not neutrophils. There are transgenic zebrafish lines to be sure about the identification of these cells. The authors could use them in order to better characterize this population.

This is a good point, which was also raised by the other reviewer. We recognize the importance of this point and have performed a very similar analysis to the one shown in figure 1. This time utilizing a double fluorescent marker line TG(*mpeg:mCherryF/mpx:GFP*) which labels macrophages and neutrophils in the same individuals. The results of this analysis have been added to the supplemental figure 1, and very clearly shows that conventionalization of zebrafish larvae is characterized by a decrease in macrophages and increase in neutrophils in the intestines.

- Which is the meaning of an increase of neutrophils and a decrease in macrophages after colonization?

This is a very good question, and one that we cannot yet give any definite answer to. But the observation is interesting in the context of other research in the field. The increase in neutrophil presence in response to conventionalization has been described before, and has been found to migrate to the intestine in a *myd88* dependent fashion in response to LPS stimulation¹⁵. To the best of our knowledge we are the first to describe the elevated intestinal macrophage presence in the germ-free state. Their function in the intestine is difficult to speculate about as we have no indication of their activation status. We feel confident that they are not classically activated, due to the absence of any detectable *il1b* expression, and we have no indication that they would be alternatively activated either. We only know that the regulation is dependent on Myd88 mediated action. We hope to address the function of the macrophages in follow-up studies and that the field will share our enthusiasm for this task. If other groups with other animal models can show similar leukocyte distribution patterns, it may emerge as an important observation to improve our understanding of the dynamics of the immune regulatory responses to early enteric colonization. In response to this comment, and that of reviewer #1 major comment 2, we have carried out an analysis of the extent to which the transcriptomics data can be related to macrophages and neutrophils in the WT and *myd88* mutant background, by comparing the lists of significantly regulated transcripts (Supplementray tables 1 and 2) to transcripts carrying either the “macrophage” or “neutrophil” descriptor at the Zfin annotation database (supplementary table 8). As briefly mentioned in the response to reviewer #1 we think an interesting candidate gene, identified from our transcriptomics data set, would be the immunoresponsive gene 1 like (*Irg1l*). This gene has previously been shown to be involved in immune migration².

- Could we speak about an intestinal inflammation in Myd88 deficient fish with the transcriptomic response and the different number of immune cells?

We do not believe it would be accurate to describe any of these conditions as inflammatory. As mentioned in figure 2 and in the manuscript text, there are hardly any detectable regulation of inflammatory mediators. In the *myd88* deficient fish the situation is much the same as in the WT in terms of inflammatory status. And only very few leukocyte-associated transcripts are regulated. One exception is the *Ita4h* involved in the conversion of leukotriene A4 into the proinflammatory eicosanoid leukotriene B4¹⁶, which shows a significant reduction in response to conventionalization only in the *myd88* mutant background. Yet as this transcriptional regulation is not manifested in the numbers of L-plasmid positive cells in the intestine (figure 1), we do not believe there is any basis for assuming an intestinal inflammatory state in the mutants, at least not one that can be recognized by the regulation of commonly held markers of inflammation.

- Why did the authors select the two bacterial strains for the monoassociated larval treatments?

This point was also raised by reviewer #1, so evidently, we have failed to convey the reasoning properly in the manuscript. For that we apologize. Our answer to these comments are the same.

Our reasoning for picking the *Exiguobacterium* and *Chryseobacterium* strains was twofold:

1) We wanted our transcriptomics data sets to be useful to the microbiome community in a broad context. These strains represent the phyla Firmicutes and Bacteroidetes which are the focus of significant research interest in the microbiome field. Upon publication these data sets could allow comparative research into the unifying features of these phyla in the context of host-microbe interactions across the various model organisms and study designs and might thus serve as a useful resource to the field at large.

2) The fact that these species are not numerically dominant in the natural situation is important to the interpretation of the comparative analysis of the transcriptomics data, because we were aiming to analyze which part of the host transcriptomic response to colonization is conserved across very different colonization conditions. Therefore, it becomes important that the mono-association data sets are both fundamentally different from each other and from the conventionalized group. That we may confidently conclude that the 65 transcripts that we found significantly regulated in all these three data sets, constitute a conserved reflection of the host response to colonization in the broadest terms.

We see this as a strength of this analysis, even though it obviously has its limitations: Another study design might have focused more on mono-association conditions that more closely resemble the composition of conventionalization conditions. We also recognize that distant from conventionalization conditions though our mono-associations may be, they are still considered commensal, and we do not know how the transcriptomic profile might differ if the mono-association had been with a pathogenic strain. As follow up studies such study designs may add further clarity. Ultimately though, any study design has its strengths and limitations. As we see it, the choice of mono-association groups allows us to define a restricted set of transcripts that robustly respond to very diverse colonization and has the potential to inform the wider field in the process of dissecting and characterizing the nature of the very complex host-microbe interactions.

We have added the following text to the manuscript to try to make these deliberations appear clearer in lines 206-210:

“Thus, the mono-associated colonization groups are likely to represent very different microbial communities to those of the conventionalized group, and any transcriptional regulation that is shared between such diverse colonization conditions is likely to be very robust to differences in the nature of colonizing microbial communities.”

- Do *myd88* deficient fish show transcriptomic differences compared with WT, both in GF conditions and after colonization? It is not clear in the manuscript. For example, in Supp. Fig 2, are these genes all the differentially expressed after colonization in *myd88*^{-/-}? Are genes related with lipid metabolism differentially expressed between WT and *myd88*^{-/-} fish?

Starting with the question of whether supplementary figure 2 of the original submission listed all the differentially expressed genes in the *myd88*^{-/-} background: the answer is no, it did not. The point of supplementary table 2 was to present the transcripts with clear connections to lipid metabolism and cholesterol. This, we now realize, did not have the intended effect of providing clarity of the effects. In this revised version a full list of the 84 transcripts that are regulated in response to conventionalization in the *myd88* deficient background, can be found as supplementary table 2. In addition to this, we have added a more unbiased approach to the functional annotation by carrying out a gene ontology (GO) analysis by the online bioinformatics resource DAVID (<https://david.ncifcrf.gov/>). Lists of enriched GO identifiers of any data set mentioned in the manuscript can be found as supplementary table 7 listing the identifiers by p-value. Furthermore, we have added a graphical representation of these lists to illustrate the difference in GO enrichment in the different data sets (supplementary figure 7). The *myd88*^{-/-} data set comprises 23 GO identifiers which very clearly demonstrates the uniform nature of the described functions of the regulated transcripts in lipid metabolism and cholesterol circulation and has no overlap with the WT data sets.

Regarding the question of whether the WT fish display transcriptional differences from *myd88* mutant fish independent of the colonization status, the answer is yes. Analyzing the differences in transcription in the *myd88* mutant and the WT fish, keeping the colonization status constant, one arrives at a much more extensive list of regulated transcripts. This reflects a profound transcriptional consequence deriving from genotype independently of colonization status.

GO analysis reveals that regulation of lipid uptake and metabolism is a very important functional category differing between mutant and WT. There is a large overlap of regulated transcripts between these genotype specific regulation patterns and those induced by colonization in the mutant background in our study (supplementary table 2). All save 2 of the transcripts listed in supplementary table 3 (supplementary table 2 of the previous submission) are differentially regulated a *myd88* genotype specific manner independently of conventionalization. This information has been added to supplementary table 3. This indicates that *myd88* serves an important function in maintaining lipid metabolic homeostasis independently of colonization status.

We agree that this function of *myd88* in relation to lipid metabolic homeostasis is very interesting, and we are planning to do metabolomics analysis to further study the physiological relevance of the transcriptomic changes. However, as further extensive analysis of the transcriptomic data does distract from the major messages of this paper, with its focus on the influence of microbiota on

myd88 regulation and host immune responses, we have chosen to not to elaborate much on this subject.

References:

1. Nguyen-Chi, M. *et al.* Identification of polarized macrophage subsets in zebrafish. *Elife* **4**, 1–14 (2015).
2. Hall, C. J. *et al.* Epidermal cells help coordinate leukocyte migration during inflammation through fatty acid-fuelled matrix metalloproteinase production. *Nat. Commun.* **5**, 3880 (2014).
3. Sun, Y. *et al.* Strategies to identify natural antisense transcripts. *Biochimie* **132**, 131–151 (2017).
4. Thisse, C. & Thisse, B. High-resolution in situ hybridization to whole-mount zebrafish embryos. *Nat. Protoc.* **3**, 59–69 (2008).
5. Hall, C. *et al.* Transgenic zebrafish reporter lines reveal conserved Toll-like receptor signaling potential in embryonic myeloid leukocytes and adult immune cell lineages. *J. Leukoc. Biol.* **85**, 751–765 (2009).
6. Li, Y.-H. *et al.* Progranulin A-mediated MET Signaling Is Essential for Liver Morphogenesis in Zebrafish. *J. Biol. Chem.* **285**, 41001–41009 (2010).
7. Medzhitov, R. *et al.* MyD88 Is an Adaptor Protein in the hToll/IL-1 Receptor Family Signaling Pathways. *Mol. Cell* **2**, 253–258 (1998).
8. Fitzgerald, K. A. *et al.* Mal (MyD88-adaptor-like) is required for Toll-like receptor-4 signal transduction. *Nature* **413**, 78–83 (2001).
9. Yang, S., Marín-juez, R., Meijer, A. H. & Spaink, H. P. Common and specific downstream signaling targets controlled by Tlr2 and Tlr5 innate immune signaling in zebrafish. *BMC Genomics* 1–10 (2015). doi:10.1186/s12864-015-1740-9
10. Mathias, J. R. *et al.* Live imaging of chronic inflammation caused by mutation of zebrafish Hai1. *J. Cell Sci.* **120**, 3372–83 (2007).
11. Rhodes, J. *et al.* Interplay of pu.1 and Gata1 determines myelo-erythroid progenitor cell fate in zebrafish. *Dev. Cell* **8**, 97–108 (2005).
12. Bhattacharyya, S., Midwood, K. S., Yin, H. & Varga, J. Toll Like Receptor-4 Signaling Drives Persistent Fibroblast Activation and Prevents Fibrosis Resolution in Scleroderma. *Adv. Wound Care* **6**, wound.2017.0732 (2017).
13. D’Arpa, P. & Leung, K. P. Toll-Like Receptor Signaling in Burn Wound Healing and Scarring. *Adv. Wound Care* **6**, 330–343 (2017).
14. Patidar, A. *et al.* DAMP-TLR-cytokine axis dictates the fate of tumor. *Cytokine* **104**, 114–123 (2017).
15. Bates, J. M., Akerlund, J., Mittge, E. & Guillemin, K. Intestinal Alkaline Phosphatase Detoxifies Lipopolysaccharide and Prevents Inflammation in Zebrafish in Response to the Gut Microbiota. *Cell Host Microbe* **2**, 371–382 (2007).

16. Chatzopoulou, A. *et al.* Glucocorticoid-induced attenuation of the inflammatory response in zebrafish. *Endocrinology* **157**, 2772–2784 (2016).

Reviewers' comments:

Reviewer #1 (Remarks to the Author):

The manuscript is much improved, and the extensive new experiments and analysis significantly strengthen the paper. I still have a few major and minor concerns that I suggest the authors address with textual changes.

Old Major Concerns:

7. Concerns over the limited characterization of the tlr2 mutant: The new data characterizing the tlr2 mutant is helpful. If available, qPCR of tlr2 mRNA in homozygous mutant and wild-type animals would be helpful to show if the transcript is subject to NMD. Also, on line 234, it is incorrect to refer to this as a "null" allele because you have not provided enough genetic evidence to show this. The current data indicate that this is a moderate/strong loss of function allele.

9. It is helpful that the authors now include the full list of significant genes from the two monoassociations. However, in light of these data, the paper has to consider the non-MCHT responses to the different microbial treatments. There is still text in the manuscript that overlooks/dismisses significantly altered genes that are NOT in the shared set of 65 MCHT genes. Several such cases are listed below, and should be addressed:

- On line 219 "This shows that most of the MCHT effects exerted by a complex microbial community can be mimicked by mono-association, an exposure that must be considered drastically different in nature from conventionalization". This statement is inaccurate. This statement is referring to the results described immediately above about the 65 shared response genes, however, for each monoassociation there were many genes not included in that set of 65 genes. So the use of the term "most" should either be replaced with "many" or the sentence should be reworked entirely.

- On line 25, replace "...is general in that it is also observed with two different..." with "... can be largely recapitulated with two different...". To say that the host response is "general" suggests that there is no specificity to the response to microbial communities or strains, which is of course not the case. Same concern with the use of the term "general" on Line 194.

- On line 157, it's inappropriate to use the term "transcriptome" here since you've only evaluated a few genes at this point in the manuscript. On the same sentence, it seems strange to infer that MCHT "could be considered as suppression of an aberrant response of the host to germ-free conditions" without considering other simple interpretations like MCHT is a shared transcriptional response to microbiome (including genes that go both up and down). The concept of "aberrant" here suggests that the colonized state is "normal". In studies such as this, it is safest to use language that discusses the phenotypes in different colonized conditions as simply different due to different types of stimuli or absence thereof, and avoid language that assigns "normality" to any one condition. Same concern with "aberrant" on Line 336 and in the title of the manuscript itself. Please consider removing the word "aberrant" from the title and these other locations.

- On line 290, "Since our dataset appears to be very robust to deviations in the composition of commensal microbial communities..." This is a strange sentence because only 65 genes out of the whole study were robust to the tested alterations in the composition of the microbial community. I could take the same data and conclude that there were abundant differences between the different community responses, and that this may indeed indicate that likely differences in baseline microbiota composition between your and other facilities is at least partially due to the differences between your dataset and published datasets such as the one you cited. Perhaps you are referring specifically to your operationally-defined MCHT set, which represents only part of the host response? If so state that explicitly.

New Major Concern:

1. Line 178: This sentence "other possible means of sensing microbial communities seems to contribute relatively little to the transcriptional regulation" is another overstatement not supported by the presented data. I might consider the same data and conclude that *myd88* mutants have such severe physiological/metabolic derangements that their host response bears almost no resemblance to WT. That would not exclude other pathways from playing important roles in host response in otherwise WT animals, but instead indicate that the *myd88* mutant simply has a strong and deranged phenotype. I suggest deleting this sentence, and avoiding such overstatements.

Minor Concerns:

I still find the use of the abbreviation/term MCHT to be unnecessary and potentially confusing to readers current and future. As you already note in the manuscript, other studies in the past (and presumably also in the future) comparing GF-CV larvae transcriptomes have seen rather different sets of genes changed. So while the MCHT term may be useful for your paper, it is quite unlikely to apply uniformly to other labs and studies. I would recommend replacing this term with something like "shared response genes".

Line 71: It would be appropriate here to include some introduction to Tlr2, as that is a major focus of the paper.

The CONVD image in Fig 1B shows apparently no L-plastin cells in the distal gut, yet the quantitation right next to it suggests each CONVD gut should have an average of ~20/gut. Please choose a representative image that reflects the mean values in the graph.

Why are some transgenic names italicized in the text/legends/figures and others not? Also, use Tg instead of TG, consistent with ZFIN guidelines.

Line 148-150: The structure of this sentence is strange and hard to understand. Also, it's not clear what they mean by "validate the absence of intestinal regulation of *il1b*". Do they mean microbial suppression of *il1b*? On that note, in Fig S4D, results from *il1b*:GFP are interesting, but showing a single animal per group without quantitation is not enough data to include in this paper. Please either show quantification of this data (either whole animal and/or gut-specific) or show multiple animals per treatment group. To me it looks like there is reduced GFP in CONV, which is not what the authors are concluding in the text.

Line 165: replace "datasets in the *myd88* mutant" with "datasets from larvae in the *myd88* mutant". This is to remind the reader that you're presenting whole-larval data.

Line 265: Replace "the larval colonization system that is defined by non-starved never-fed..." with "the larval colonization system we used that is defined by never-fed...". This edit is to make it clear these are the methods you selected (not necessarily a standard in the field) and also removed the unnecessary non-starved statement.

Line 269: Replace "in the presence of an adaptive immune system" with "in the presence of an adaptive immune system and feeding".

Line 285: This sentence is an odd overstatement, is unsupported by the data, and must be deleted : "...which seems to be at the heart of MCHT"

Line 308: This may be a useful point to reference recent work in zebrafish that has defined at least

one transcriptional pathway that appears to mediate microbiota suppression of GI genes including the same metabolic pathways you are referring to here (see PMID PMC5495071).

Line 390: Clarify if culturing was done under aerobic and/or anaerobic conditions.

Reviewer #2 (Remarks to the Author):

I think that the authors have explained very well all the improvements conducted in the manuscript in the rebuttal letter. I really think that now the results are clearer and suitable for publication.

We are very grateful for the insightful comments. We address each reviewer comment below. Our answers appear in blue font.

Reviewers' comments:

Reviewer #1 (Remarks to the Author):

The manuscript is much improved, and the extensive new experiments and analysis significantly strengthen the paper. I still have a few major and minor concerns that I suggest the authors address with textual changes.

Old Major Concerns:

7. Concerns over the limited characterization of the *tlr2* mutant: The new data characterizing the *tlr2* mutant is helpful. If available, qPCR of *tlr2* mRNA in homozygous mutant and wild-type animals would be helpful to show if the transcript is subject to NMD. Also, on line 234, it is incorrect to refer to this as a “null” allele because you have not provided enough genetic evidence to show this. The current data indicate that this is a moderate/strong loss of function allele.

We agree with the reviewer that it is difficult to unambiguously demonstrate that a point mutation is a complete null mutant. Therefore, we have changed the statement in the text (lines 230-234) to read:

“To investigate whether Tlr2 could be involved in the microbial suppression of *myd88* transcription, we performed qPCR analysis to assess transcription of *myd88* and two other shared transcriptional response markers in conventionalized and germ-free larvae of Tlr2 loss of function mutant fish line (*tlr2*^{sa19423}, see materials and methods) compared to wildtype siblings (figure 4 C).”

We have tried RT-PCR, but needed nested primers to detect the mRNA, which we did not consider reliable for quantitation. However, we have done RNAseq experiments in *tlr2* mutant and WT controls which, although the read counts are very low, indicated no difference between WT and mutants, indicating that the transcript is not subject to nonsense mediated decay. A figure showing the mapped read counts to *tlr2* mutant versus WT controls have been added as supplementary figure 8, and reference to this supplemental figure has been added to the materials and methods. A more detailed analysis of the RNAseq data of the *tlr2* mutant and its responses to mycobacterial infection will follow in another manuscript which is currently in preparation.

Note added in final revision: Following the editors' suggestion supplementary figure 8 has been removed during final revisions, but will feature in the review correspondence. The mentioning of this supplementary figure has been removed from the manuscript.

Supplementary Figure 8. RNAseq read counts of *tlr2* transcripts in ABTL control versus *tlr2* mutant larvae.

RNAseq data comparing reads mapped to the *tlr2* transcript (ENSDART0000012256). Similarly to the WT mutant reads are mapped to the entire length of the mutant transcript indicating that the mutant transcript is not subjected to nonsense mediated decay. The figure is based on mutant data (**MEAS**) that has been submitted to the NCBI gene expression Omnibus database, accession number (GSE102766), WT (**CTRL**) data was extracted from a previously published dataset⁵⁴ (mutants, n=3 biological replicates, 10 embryos per group, WT, n=4 biological replicates, n ~100 embryos per group).

9. It is helpful that the authors now include the full list of significant genes from the two monoassociations. However, in light of these data, the paper has to consider the non-MCHT responses to the different microbial treatments. There is still text in the manuscript that overlooks/dismisses significantly altered genes that are NOT in the shared set of 65 MCHT genes. Several such cases are listed below, and should be addressed:

- On line 219 “This shows that most of the MCHT effects exerted by a complex microbial community can be mimicked by mono-association, an exposure that must be considered drastically different in nature from conventionalization”. This statement is inaccurate. This statement is referring to the results described immediately above about the 65 shared response genes, however, for each monoassociation there were many genes not included in that set of 65 genes. So the use of the term “most” should either be replaced with “many” or the sentence should be reworked entirely.

This is valuable input.

We have altered the sentence accordingly, and it now reads:

Lines 217-219

“This shows that many of the transcriptional effects regulated by colonization by a complex microbial community can be mimicked by mono-association, an exposure that must be considered drastically different in nature from conventionalization”.

- On line 25, replace "...is general in that it is also observed with two different..." with "... can be largely recapitulated with two different...". To say that the host response is "general" suggests that there is no specificity to the response to microbial communities or strains, which is of course not the case. Same concern with the use of the term "general" on Line 194.

We are grateful for the input and for the suggestions for improvement and clarification. We have changed the sentence in the abstract (line 25-26) according to the suggestion, so that it reads:

"Mono-association studies show that the microbiome control of the host transcriptome can be largely recapitulated with two different commensal strains from zebrafish."

In line 194 (now line 192) we have substituted the word "general" to "conserved" so that the line now reads:

"Gnotobiotic treatment show conserved aspects of microbiome control of the host transcriptome"

- On line 157, it's inappropriate to use the term "transcriptome" here since you've only evaluated a few genes at this point in the manuscript. On the same sentence, it seems strange to infer that MCHT "could be considered as suppression of an aberrant response of the host to germ-free conditions" without considering other simple interpretations like MCHT is a shared transcriptional response to microbiome (including genes that go both up and down). The concept of "aberrant" here suggests that the colonized state is "normal". In studies such as this, it is safest to use language that discusses the phenotypes in different colonized conditions as simply different due to different types of stimuli or absence thereof, and avoid language that assigns "normality" to any one condition. Same concern with "aberrant" on Line 336 and in the title of the manuscript itself. Please consider removing the word "aberrant" from the title and these other locations.

The point was that a germ-free state is far removed from "normal" and that was why we used the term aberrant. However, we recognize the concern and have removed the term aberrant from the manuscript all together.

The summary section in lines 153-158 now reads:

"In summary, intestinal colonization affects the intestinal immune status in several ways: It leads to an overall decrease in leukocyte presence, marked by a decrease of macrophages but a rise in neutrophil presence, these changes of leukocyte populations are dependent on Myd88 mediated signaling. On a transcriptional level the effects of colonization are characterized by a suppression of myd88 and downstream regulated genes in the intestine, but absence of pro-inflammatory il1b regulation."

- On line 290, "Since our dataset appears to be very robust to deviations in the composition of commensal microbial communities..." This is a strange sentence because only 65 genes out of the whole study were robust to the tested alterations in the composition of the microbial community. I could take the same data and conclude that there were abundant differences between the different community responses, and that this may indeed indicate that likely differences in baseline microbiota composition between your and other facilities is at least partially due to the differences

between your dataset and published datasets such as the one you cited. Perhaps you are referring specifically to your operationally-defined MCHT set, which represents only part of the host response? If so state that explicitly.

We appreciate this comment. We have altered the passage in the manuscript to make it clear that we are addressing the lack of overlap between our shared response gene set (we have removed the term MCHT), and to make the sentence less strong, by acknowledging that some of the differences between our data and the Kanther study may arise from differences in microbial community composition. The passage (lines 286-294) now reads:

“Interestingly, a microarray transcriptome study of germ-free versus conventionalized conditions by Kanther et al.¹⁰, a study in which the larval host had received feeding and analysis took place at 6 DPF, revealed little overlap of regulated transcripts with our shared response transcript set. Since our shared response gene set appears to be very robust to deviations in the composition of commensal microbial communities, it seems unlikely that all the differences are caused by the differences in microbial communities in the conventionalization protocol. Several other possible reasons might add to explaining these differences, among others, circadian regulation, zebrafish genetic strains or the introduction of feeding in the system.”

New Major Concern:

1. Line 178: This sentence “other possible means of sensing microbial communities seems to contribute relatively little to the transcriptional regulation” is another overstatement not supported by the presented data. I might consider the same data and conclude that *myd88* mutants have such severe physiological/metabolic derangements that their host response bears almost no resemblance to WT. That would not exclude other pathways from playing important roles in host response in otherwise WT animals, but instead indicate that the *myd88* mutant simply has a strong and deranged phenotype. I suggest deleting this sentence, and avoiding such overstatements.

This is a good point. We have removed the sentence.

Minor Concerns:

I still find the use of the abbreviation/term MCHT to be unnecessary and potentially confusing to readers current and future. As you already note in the manuscript, other studies in the past (and presumably also in the future) comparing GF-CV larvae transcriptomes have seen rather different sets of genes changed. So while the MCHT term may be useful for your paper, it is quite unlikely to apply uniformly to other labs and studies. I would recommend replacing this term with something like “shared response genes”.

We appreciate this concern. We have removed the term MCHT from the manuscript, and replaced it with various descriptions dependent on the context:

For each occurrence the previous wording is shown in red and the current in blue (line designations refer to the current manuscript).

Lines 163-164:

“Considering that transcription of *myd88* itself is suppressed by conventionalization, we investigated its role in MCHT.”

“Considering that transcription of *myd88* itself is suppressed by conventionalization, we investigated its role in the colonization sensitive transcriptional regulation in the host.”

Lines 175-177:

Thus, nearly the entire MCHT signature seems dependent on Myd88 signaling (supplementary figure 6).

“Thus, nearly the entire microbiome control of the host transcription signature seems dependent on Myd88 signaling (supplementary figure 6).”

Lines 217-219:

“This shows that many of the MCHT effects exerted by a complex microbial community can be mimicked by mono-association, an exposure that must be considered drastically different in nature from conventionalization.”

“This shows that many of the transcriptional effects regulated by colonization by a complex microbial community can be mimicked by mono-association, an exposure that must be considered drastically different in nature from conventionalization.”

Lines 219-221:

“The identified set of 65 shared genes can be used as a common marker set for MCHT which seems independent of the nature or treatment dose of the stimulating microbiota.”

“The identified set of 65 shared genes can be used as a common marker set of shared transcriptional response which seems independent of the nature or treatment dose of the stimulating microbiota.”

Lines 224-229 (two exchanges):

“While analyzing the MCHT marker set (figure 3 C) we identified a large overlap with a set of 48 transcripts constituting the TLR2 dependent response to the injection of the synthetic ligand Pam3CSK4 which mimics the bacterial lipopeptide³¹ (see figure 4 A). Out of the 65 MCHT transcripts, 11 were shared with the Tlr2 dependent transcription response set”

“While analyzing the shared transcriptional response marker set (figure 3 C) we identified a large overlap with a set of 48 transcripts constituting the TLR2 dependent response to the injection of the synthetic ligand Pam3CSK4 which mimics the bacterial lipopeptide³¹ (see figure 4 A). Out of the 65 shared response transcripts, 11 were shared with the Tlr2 dependent transcription response set”

Lines 230-234:

“To investigate whether Tlr2 could be involved in the microbial suppression of *myd88* transcription, we performed qPCR analysis to assess transcription of *myd88* and two other MCHT markers in

conventionalized and germ-free larvae of Tlr2 loss of function mutant fish line (*tlr2*^{sa19423}, see materials and methods) compared to wildtype siblings (figure 4 C).”

“To investigate whether Tlr2 could be involved in the microbial suppression of *myd88* transcription, we performed qPCR analysis to assess transcription of *myd88* and two other shared transcriptional response markers in conventionalized and germ-free larvae of Tlr2 loss of function mutant fish line (*tlr2*^{sa19423}, see materials and methods) compared to wildtype siblings (figure 4 C).”

Lines 236-237:

“The downregulation of other markers is lower and no longer significant for the other selected markers of MCHT *cebpb* and *fosl1a* in the Tlr2 mutant.”

“The regulation no longer significant for *cebpb* and *fosl1a*, the other selected markers of shared transcriptional response, in the Tlr2 mutant.”

Lines 260-265 (4 exchanges):

“Having defined a gene marker set of microbiome control of the host transcriptome (MCHT) and considering that suppressed *myd88* expression is a signature of this control, we investigated the function of Myd88 in MCHT. Results from *myd88* deficient mutants revealed that Myd88 signaling is indispensable for MCHT. This dependence of MCHT on Myd88 could reflect the simplicity of the larval colonization system we used that is defined by never-fed conditions eliminating a number of confounding factors.”

“Having defined a gene marker set of microbiome control of the host transcriptome and considering that suppressed *myd88* expression is a signature of this control, we investigated the function of Myd88 in the microbiome sensitive transcriptional regulation. Results from *myd88* deficient mutants revealed that Myd88 signaling is indispensable for the regulation. This dependence on Myd88 could reflect the simplicity of the larval colonization system we used that is defined by never-fed conditions eliminating a number of confounding factors.”

Lines 273-277 (2 exchanges):

“Using a gnotobiotic system we investigated whether the identified gene marker set for MCHT is dependent on the microbiome characteristics. The MCHT gene sets are remarkably similar and, in most cases, the same genes are regulated in the same direction when comparing two mono-associations with conventionalized conditions.”

“Using a gnotobiotic system we investigated whether the identified gene marker set for microbiome control of host transcriptome is dependent on the microbiome characteristics. The transcriptional regulation is remarkably similar, and, in most cases, the same genes are regulated in the same direction when comparing two mono-associations with conventionalized conditions.”

Line:

“This generic set included *myd88* along with several of the other signature transcripts of MCHT (supplementary table 1 and figure 4D), indicating that microbiome regulation of *myd88* transcription, which seems to be at the heart of MCHT, can also be exerted by both of the species of commensal bacteria tested in this study.”

“This generic set included *myd88* along with several of the other signature transcripts of shared transcriptional response (supplementary table 1 and figure 4D), indicating that microbiome regulation of *myd88* transcription, can also be exerted by both of the species of commensal bacteria tested in this study.”

Lines 283-286:

“The 11 overlapping transcripts between the 65 MCHT markers and the 48 genes constituting the TLR2 dependent response to PAM3CSK4 (figure 4)....”

“The 11 overlapping transcripts between the 65 shared transcriptional response markers and the 48 genes constituting the TLR2 dependent response to PAM3CSK4 (figure 4)....”

Lines 735-736 (figure label 3):

“These 65 genes represent strong candidates for markers of MCHT.”

“These 65 genes represent strong candidates for markers of the shared transcriptional response.”

Lines 741-745 (figure label 4):

“**A:** Venn diagram showing the overlap between the identities of 65 primarily suppressed transcripts displaying transcriptional sensitivity to all the different colonization modes tested in this study, and thus are strong markers of MCHT, with 48 primarily induced transcripts displaying TLR2 dependent transcriptional sensitivity to injection of the synthetic ligand Pam3CSK4.”

“**A:** Venn diagram showing the overlap between the identities of 65 primarily suppressed transcripts displaying transcriptional sensitivity to all the different colonization modes tested in this study, and thus are strong markers of the shared transcriptional response, with 48 primarily induced transcripts displaying TLR2 dependent transcriptional sensitivity to injection of the synthetic ligand Pam3CSK4.”

Line 71: It would be appropriate here to include some introduction to Tlr2, as that is a major focus of the paper.

This is a good point. We have added the following lines to the introduction (lines 46-50)

“Especially TLR2 has attracted interest in this regard for several reasons. TLR2 mediated microbial pattern recognition has been shown to be important for facilitating tolerance to commensal microbial colonization¹¹, induction of mucin secretion¹² and protection of intestinal barrier integrity in induced inflammation models^{13,14}.”

The CONVD image in Fig 1B shows apparently no L-plastin cells in the distal gut, yet the quantitation right next to it suggests each CONVD gut should have an average of ~20/gut. Please choose a representative image that reflects the mean values in the graph.

The image has been replaced by a more representative one.

Why are some transgenic names italicized in the text/legends/figures and others not? Also, use Tg instead of TG, consistent with ZFIN guidelines.

These mistakes have been corrected.

Line 148-150: The structure of this sentence is strange and hard to understand. Also, it's not clear what they mean by "validate the absence of intestinal regulation of *il1b*". Do they mean microbial suppression of *il1b*? On that note, in Fig S4D, results from *il1b*:GFP are interesting, but showing a single animal per group without quantitation is not enough data to include in this paper. Please either show quantification of this data (either whole animal and/or gut-specific) or show multiple animals per treatment group. To me it looks like there is reduced GFP in CONV, which is not what the authors are concluding in the text.

The message that the text was supposed to convey was this: transcriptome data had revealed no significant expressional regulation of *il1b*, and the tissue specific qPCR data in S4D was a validation that there was also no detectable *il1b* response in the intestine. We are grateful to the reviewer for pointing out that this meaning was not obvious to an outside reader. The sentence (lines 151-152) has been altered so that it now reads:

"The tissue specific qPCR approach further validated that *il1b* is also not significantly regulated in the intestine (supplementary figure 4)."

Regarding figure S4D we appreciate the reviewers concern that showing one animal does not convey much information regarding variability. The images in figure S4D have been replaced to depict three larvae per treatment, in a manner that more faithfully represents the variability within each group.

Line 165: replace "datasets in the *myd88* mutant" with "datasets from larvae in the *myd88* mutant". This is to remind the reader that you're presenting whole-larval data.

We have followed the advice and altered the sentence (lines 165-166).

Line 265: Replace "the larval colonization system that is defined by non-starved never-fed..." with "the larval colonization system we used that is defined by never-fed...". This edit is to make it clear these are the methods you selected (not necessarily a standard in the field) and also removed the unnecessary non-starved statement.

We have followed the advice and altered the sentence (lines 264-265).

Line 269: Replace “in the presence of an adaptive immune system” with “in the presence of an adaptive immune system and feeding”.

We have followed the advice and altered the sentence.

Line 285: This sentence is an odd overstatement, is unsupported by the data, and must be deleted : “...which seems to be at the heart of MCHT”

We have followed the advice and altered the sentence (lines 268-269).

Line 308: This may be a useful point to reference recent work in zebrafish that has defined at least one transcriptional pathway that appears to mediate microbiota suppression of GI genes including the same metabolic pathways you are referring to here (see PMID PMC5495071).

We thank the reviewer for making us aware of this interesting parallel. We have added the following passage to the discussion (lines 306-311):

“Interestingly many genes of these metabolic pathways affected by colonization in the *myd88* mutant were recently found to be affected by colonization in mutants of the *hnf4a* gene, which specifically binds and activates a microbiota suppressed intestinal epithelial transcriptional enhancer.³⁶ It will be interesting to study whether there is a functional link between *myd88* mediated microbial recognition and the *Hnf4* gene, and its link with inflammatory bowel diseases as shown by Davison *et al*³⁶.”

Line 390: Clarify if culturing was done under aerobic and/or anaerobic conditions.

It was under aerobic conditions. The clarification has been added (line 398).

Reviewer #2 (Remarks to the Author):

I think that the authors have explained very well all the improvements conducted in the manuscript in the rebuttal letter. I really think that now the results are clearer and suitable for publication.